# Mapping ticks and tick-borne pathogens in China

Guo-Ping Zhao [1,2], Yi-Xing Wang [1], Zheng-Wei Fan [1], Yang Ji[1], Ming-jin Liu[3], Wen-Hui Zhang [1], Xin-Lou Li[1], Shi-Xia Zhou [1], Hao Li[1], Song Liang [3], Wei Liu [1✉], Yang Yang [3✉] & Li-Qun Fang [1✉]

Understanding ecological niches of major tick species and prevalent tick-borne pathogens is crucial for efficient surveillance and control of tick-borne diseases. Here we provide an up-to-date review on the spatial distributions of ticks and tick-borne pathogens in China. We map at the county level 124 tick species, 103 tick-borne agents, and human cases infected with 29 species (subspecies) of tick-borne pathogens that were reported in China during 1950−2018. *Haemaphysalis longicornis* is found to harbor the highest variety of tick-borne agents, followed by *Ixodes persulcatus*, *Dermacentor nutalli* and *Rhipicephalus microplus*. Using a machine learning algorithm, we assess ecoclimatic and socioenvironmental drivers for the distributions of 19 predominant vector ticks and two tick-borne pathogens associated with the highest disease burden. The model-predicted suitable habitats for the 19 tick species are 14–476% larger in size than the geographic areas where these species were detected, indicating severe under-detection. Tick species harboring pathogens of imminent threats to public health should be prioritized for more active field surveillance.

[1] State Key Laboratory of Pathogen and Biosecurity, Beijing Institute of Microbiology and Epidemiology, Beijing, P.R. China. [2] Logistics College of Chinese People's Armed Police Forces, Tianjin, P.R. China. [3] College of Public Health and Health Professions and Emerging Pathogens Institute, University of Florida, Gainesville, FL, USA. ✉email: liuwei@bmi.ac.cn; yangyang@ufl.edu; fang_lq@163.com

Ticks are hematophagous arthropods that parasitize verte-brates including livestock, wild animals, and human beings throughout the world[1]. The major threat that ticks impose on human and animal health lies in their vector role in the transmission of a variety of pathogens, and their epidemiological and epizootic significance is considered only second to mosquitoes[2,3]. Pathogenic organisms harbored by ticks mainly encompasses viruses, bacteria (in particular rickettsiae and spir-ochetes), protozoa, and helminth, with increasing diversity over the past 30 years[4,5]. The ongoing geographic expansion of tick species, possibly driven by climatic and environmental changes, has drawn global attention[6]. A typical example is *Haemaphysalis* (*Ha.*) *longicornis*, which was originally native to East Asia, spread to Australia, New Zealand, and several Pacific Islands since 1983[7], and was recently found in the eastern U.S. The expansion of *Ha. longicornis* has raised public health and animal health concerns due to its capability of transmitting tick-borne agents, e.g., severe fever with thrombocytopenia syndrome virus (SFTSV), spotted fever group rickettsiae, and *A. phagocytophilum*[8–11]. Another example is the increasing incidence of tick-borne encephalitis (TBE) across the Euro–Asia in the past three decades, which was linked to the expansion of its competent vectors, *Ixodes* (*I.*) *ricinus* and *I. persulcatus* ticks[12,13]. This globalization trend of ticks and increasing variety of tick-borne diseases (TBD) are calling for an extensive and in-depth research on the spatial distributions of both ticks and tick-borne pathogens, as well as their underlying risk determinants.

In China, the growing awareness of emerging tick-borne pathogens has greatly inspired investigations on ticks and TBDs in recent years[4,14]. One study compiled a data set with regard to tick distribution and diversity up to the county level in China from peer-reviewed literature published between 1960 and 2017[15]. Another study reviewed the geographic distribution of tick species at the province level together with the diversity and specificity of animal hosts of ticks[16]. Yu et al.[17] reviewed the association between pathogenic microorganisms and tick vectors throughout China based on the literature up to 2014. However, none of the studies provided high resolution spatial distribution of tick-borne pathogens, nor did they investigate systematically ecological niches of either major tick species or prevalent tick-borne pathogens.

Here we conduct an up-to-date review on the spatial distribution of predominant tick species, tick-borne agents, and human cases of TBDs in China, based on which we build predictive models to assess the contributions of relevant socio-environmental factors to the ecological suitability of selected 19 ticks and two tick-borne pathogens, and map model-projected risks to inform future surveillance and control efforts. *Ha. longicornis* is found to harbor the highest variety of tick-borne agents, followed by *Ixodes persulcatus*, *Dermacentor nutalli* and *Rhipicephalus microplus*. The top five tick-borne agents that parasitize the largest number of tick species are *Anaplasma phagocytophilum*, *Borrelia burgdorferi* sensu stricto, *Borrelia garinii*, *Ehrlichia chaffeensis*, and *Rickettsia raoultii*. The model predicted suitable habitats for the 19 tick species are extensive, 14–476% larger in size than the geographic areas where these species have been observed. Tick species that are severely underdetected but harboring pathogens of imminent threats to public health should be prioritized for field surveillance.

## Results

### Distribution of tick species in mainland China.
We compiled a database comprising 7344 unique records on geographic distributions of 124 known tick species, including 113 hard tick species in seven genera and 11 soft tick species in two genera,

together with 103 tick-associated agents detected in either ticks or humans, which were recorded in 1134 counties (39% of all counties in the mainland of China) (Supplementary Fig. 1 and Supplementary Note 1). The most widely distributed tick genus was *Dermacentor* (in 574 counties), followed by *Heamaphysalis* (570), *Ixodes* (432), *Rhipicephalus* (431), *Hyalomma* (298), *Argas* (90), *Ornithodoros* (38), *Amblyomma* (37), and *Anom-alohimalaya* (5) (Supplementary Data 1 and Supplementary Figs. 2–10). At the species level, *D. nuttalli*, *Ha. longicornis*, *D. silvarum*, *Hy. scupense*, and *R. sanguineus* were each found in >200 counties, followed by *R. microplus*, *I. persulcatus*, *I. sinensis*, *I. granulatus,* and *Hy. asiaticum* that were each detected in 100–200 counties (Supplementary Data 1). We identified 19 pre-dominant ticks that were detected in more than 40 counties, including five Ixodes species, four Heamaphysalis, four Derma-centor, three Rhipicephalus, two Hyalomma, and one Argas tick species. Forest and meadowlands are the major vegetation types for these 19 tick species, accounting for a median of 46.4% (IQR: 40.0%–68.9%) of their habitats (Supplementary Data 1).

The abundance of tick species varies substantially across the seven biogeographic zones which are defined by climatic and ecological characteristics (Fig. 1)[18,19]. Tick species are most abundant in Central China, South China, and Inner Mongolia–Xinjiang districts, hosting 61, 57, and 50 tick species, respectively (Supplementary Data 2). Eight prefectures reported ≥20 tick species, three in Xinjiang Autonomous Region of northwestern China, two in Yunnan Province of southwestern China, and one in each of Gansu, Hubei, and Fujian provinces of northwestern, central, and southeastern China, respectively (Fig. 1). Most genera except for *Amblyomma* were found in northwestern China, particularly in Xinjiang Autonomous Region. In contrast, less tick diversity was observed in north-eastern China, which only harbors *Ixodes*, *Heamaphysalis,* and *Dermacentor* (Supplementary Figs. 2–10).

### Risk mapping and risk factors for 19 predominant tick species.
The ecological modeling results for the 19 predominant tick species showed highly accurate predictions, with the average testing area-under-curve (AUC) ranging from 0.83 to 0.97 (Table 1) and the testing partial AUC ratio ranging from 1.30 to 1.78 (Supplementary Tables 1–5), indicating decent predictive power. The ecoclimatic and environmental variables that were predictive of the geographic distribution of the ticks differed among the species, even for those in the same genus (Fig. 2f, Supplementary Tables 1–5). Temperature seasonality and mean temperature in the driest quarter were the two most important drivers, contributing ≥5% to the ensemble of models for 14- and 12- tick species, respectively, followed by elevation contributing ≥5% to the models for seven tick species (Fig. 2f, Supplementary Tables 1–5). The same predictor, however, may drive the risk in different directions for different tick species (Supplementary Figs. 11–29). For example, a high temperature in the driest quarter was associated with a high probability of presence for *I. granulatus* and *R. haemaphysaloide* but with a low probability for *I. persul-catus* and *Ha. longicornis* (Supplementary Figs. 11, 13, 16, 22).

The model-predicted high-risk areas of the 19 tick species were much more extensive than have been observed, 31–520% greater in the number of affected counties, 14–476% larger in the size of affected geographic area, and 25–556% larger in the affected population size (Table 1, Supplementary Figs. 30–34). *Ha. longicornis* was predicted to have the widest distribution that potentially affected 588 million people in 1140 counties, followed by *I. sinensis* and *R. microplus* that affected 363 and 350 million people in 630 and 678 counties, respectively (Table 1). High-risk areas of these three tick species collectively covered nearly all

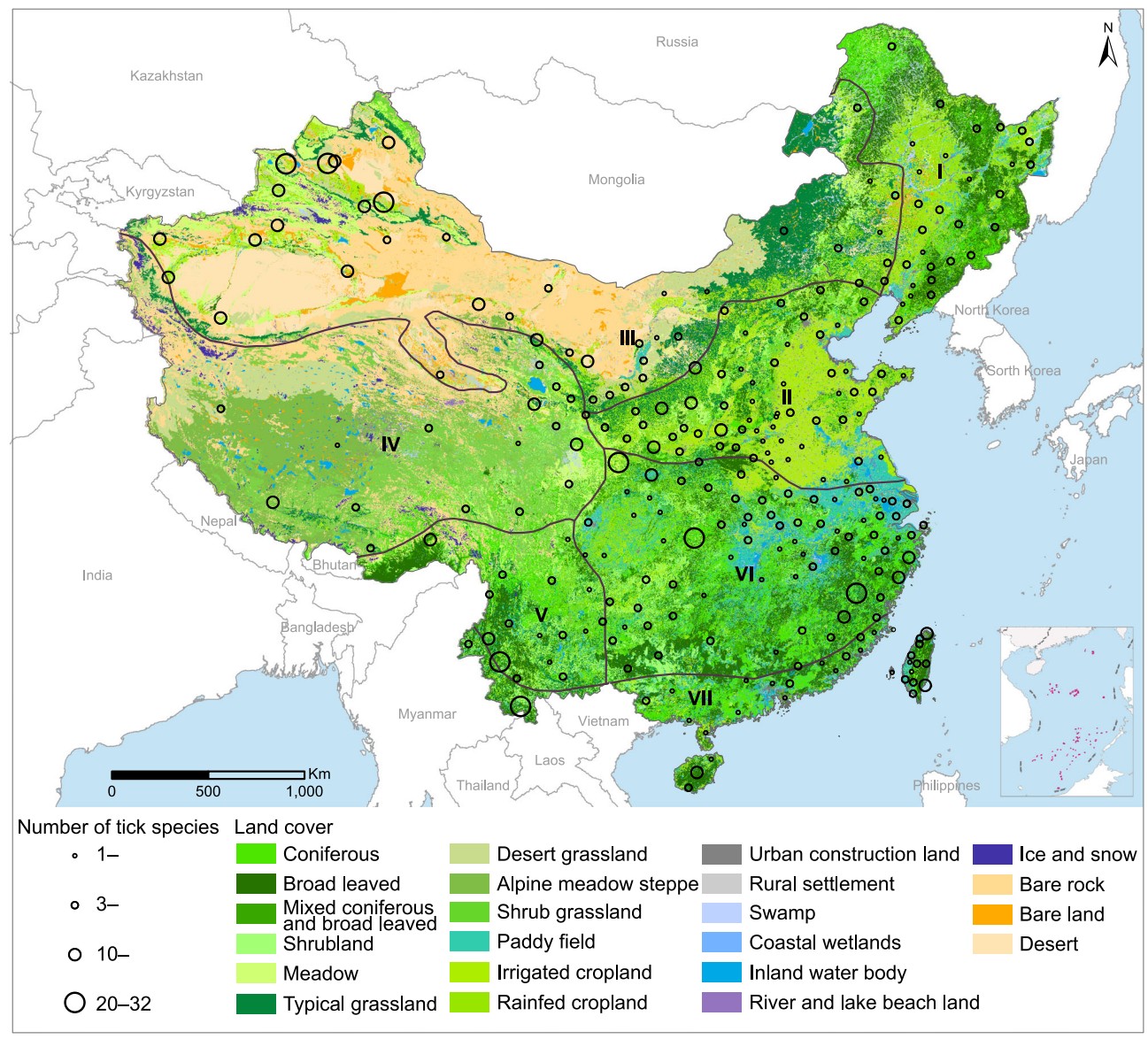

**Fig. 1 Tick species richness (circles) at the prefecture level in seven biogeographic zones in mainland China from 1950 to 2018.** I = Northeast district (NE), II = North China district (N), III = Inner Mongolia–Xinjiang district (IMX), IV = Qinghai–Tibet district (QT), V = Southwest district (SW), VI = Central China district (C), and VII = South China district (S). Source data are provided as a Source Data file.

densely populated areas in China, mainly provinces in the central, eastern, southern, and southwestern China (Supplementary Figs. 30(b), 31(a), and 32(b)). *R. sanguineus*, and *R. haemaphysaloides* each affected more than 200 million people. *D. nuttalli*, *I. crenulatus*, *Hy. asiaticum*, *Ar. persicus,* and *D. daghestanicus* ticks were the top five tick species affecting the largest areas at the scale of 2.0–3.8 million km$^2$ (Table 1).

**Ecological clustering of tick species.** Based on the ecological similarity represented by the environmental and ecoclimatic predictors, the 19 tick species were grouped into five clusters with clear patterns of spatial aggregation (Fig. 2). *D. nuttalli* and *D. silvarum* constituted cluster I that covered the vast region in northern (including northeastern and northwestern) China. This cluster stretches over biogeographic zones I–IV characterized by middle to high elevations, shrub grassland, strong seasonality in temperature, relatively low temperature in the wettest quarter (often also the warmest quarter), and low precipitation in the

driest month (Fig. 2 and Supplementary Figs. 23, 24). *Ha. longicornis*, *Hy. scupense*, and *R. sanguineus* were grouped into Cluster II which was mainly found in biogeographic zones II, III, and VI, featuring the landscape of shrub grassland and irrigated or rainfed croplands at low-middle elevations (<1600 m) in central, eastern, and northwestern China (Fig. 2 and Supplementary Figs. 16, 20, 27). *R. microplus*, *R. haemaphysaloides*, *I. granulatus* and *Ha. hystricis* were grouped into Cluster III that was mainly distributed in biogeographic zones V–VII covered by coniferous or broad-leaved woods at low elevations in southern and central China where the weather is warm and humid with low seasonality in temperature (Fig. 2 and Supplementary Figs. 13, 19, 21, 22). Cluster IV, composed of *I. persulcatus*, *Ha. concinna* and *Ha. japonica*, ecologically fits biogeographic zones I and III in northwestern and northeastern China covered by coniferous or broad-leaved forests as well as cropland, featuring strong seasonality in temperature and low temperatures in the driest season (Fig. 2 and Supplementary Figs. 11, 17, 18). Cluster V comprises

**Table 1 The average testing areas under the curve (AUC) of the BRT models at the county level and model-predicted numbers, areas and population sizes of affected counties for the 19 most prevalent tick species in China.**

| Tick species | Average AUC (2.5–97.5% percentiles) | Predicted/observed (relative difference %) | | |
|---|---|---|---|---|
| | | No. of counties | Area (10,000 km²) | Population size (million) |
| *Ixodes persulcatus* | 0.892 (0.855-0.929) | 380/135 (181.5) | 173.2/87.9 (97.0) | 149.1/47.5 (213.9) |
| *I. sinensis*[ac] | 0.924 (0.894-0.953) | 630/113 (457.5) | 81.2/14.1 (475.9) | 363.3/77.2 (370.6) |
| *I. granulatus* | 0.943 (0.920-0.965) | 373/103 (262.1) | 62.5/18.6 (236.0) | 201.4/51.5 (291.1) |
| *I. crenulatus*[b] | 0.918 (0.885-0.951) | 205/76 (169.7) | 322.3/113.0 (185.2) | 33.8/16.4 (106.1) |
| *I. ovatus* | 0.828 (0.778-0.878) | 335/57 (487.7) | 145.1/28.2 (414.5) | 110.5/20.9 (428.7) |
| *Haemaphysalis longicornis*[ac] | 0.900 (0.879-0.920) | 1140/303 (276.2) | 172.8/70.9 (143.7) | 588.0/138.3 (325.2) |
| *Ha. concinna* | 0.875 (0.827-0.922) | 165/72 (129.2) | 86.2/55.3 (55.9) | 51.1/24.9 (105.2) |
| *Ha. japonica* | 0.888 (0.835-0.941) | 95/52 (82.7) | 36.4/21.4 (70.1) | 27.7/17.8 (55.6) |
| *Ha. hystricis* | 0.893 (0.854-0.933) | 205/43 (376.7) | 43.2/12.5 (245.6) | 86.2/19.6 (339.8) |
| *Rhipicephalus sanguineus*[c] | 0.833 (0.800-0.866) | 470/221 (112.7) | 163.1/94.0 (73.5) | 236.4/96.9 (144.0) |
| *R. microplus*[ac] | 0.871 (0.844-0.898) | 678/188 (260.6) | 138.3/54.1 (155.6) | 349.6/106.8 (227.3) |
| *R. haemaphysaloides*[c] | 0.901 (0.862-0.940) | 471/76 (519.7) | 76.3/14.4 (429.9) | 232.9/35.5 (556.1) |
| *Dermacentor nuttalli*[ab] | 0.966 (0.956-0.975) | 557/382 (45.8) | 378.6/274.1 (38.1) | 153.2/109.3 (40.2) |
| *D. silvarum*[a] | 0.926 (0.907-0.944) | 492/274 (79.6) | 155.6/129.7 (20.0) | 168.1/96.2 (74.7) |
| *D. daghestanicus*[b] | 0.930 (0.882-0.977) | 112/62 (80.6) | 196.5/95.9 (104.9) | 22.0/13.6 (61.8) |
| *D. marginatus* | 0.960 (0.942-0.978) | 81/62 (30.6) | 54.3/47.7 (13.8) | 19.0/13.9 (36.7) |
| *Hyalomma scupense* | 0.912 (0.893-0.932) | 410/234 (75.2) | 168.6/137.3 (22.8) | 160.7/81.9 (96.2) |
| *Hy. asiaticum*[b] | 0.948 (0.926-0.969) | 186/135 (37.8) | 299.7/212.2 (41.2) | 39.2/31.4 (24.8) |
| *Argas persicus*[b] | 0.874 (0.836-0.902) | 225/68 (230.9) | 208.7/45.7 (356.7) | 104.2/42.3 (146.3) |

The predicted numbers are compared with the actual observations from field surveys and the relative differences (%) are given in parentheses.
[a]Top 5 tick species affecting largest numbers of counties.
[b]Top 5 tick species affecting largest areas.
[c]Top 5 tick species affecting largest population sizes.

of *Hy. asiaticum*, *D. marginatus*, *D. daghestanicus* and *Ar. persicus*, the natural habitats of which are meadow, desert grassland and cropland in biogeographic zone III as well as part of zone IV in northern and northwestern China, featuring low precipitations in the wettest/warmest quarter or month (Fig. 2 and Supplementary Figs. 25, 26, 28). Three tick species, *I. ovatus*, *I. sinensis* and *I. crenulatus*, have their own unique ecological niches and are thus not clustered with others. In terms of geographic distribution, however, *I. crenulatus* is similar to Cluster V, and *I. ovatus* and *I. sinensis* are similar to Cluster III (Fig. 2 and Supplementary Fig. 2).

**Distribution of tick-borne agents.** Among the 103 tick-borne agents detected in China, 65 were newly identified in the past two decades (Fig. 3). *Ha. longicornis* is the tick species harboring the highest variety of tick-borne agents, as many as 44 known species including seven *Rickettsia* species, seven *Babesia*, 12 *Anaplasmataceae*, four *Theileria*, four *Borrelia*, nine viruses, and *Francisella tularensis* (*F. tularensis*) (Fig. 3). Other competent tick species that carry 20 or more agents are *I. persulcatus* (36 agents), *D. nutalli* (32 agents), *R. microplus* (31), *D. silvarum* (30), *Ha. concinna* (24), and *Hy. asiaticum* (23). Agents that parasitize more than ten tick species are *R. raoultii* (in 15 tick species), *R. heilongjiangensis* (14), *Anaplasma* (*A.*) *phagocytophilum* (22), *Ehrlichia* (*E.*) *chaffeensis* (16), *A. bovis* (ten), *B. burgdorferi* sensu stricto (20), *B. garinii* (18), *B. afzelii* (11), *Coxiella* (*C.*) *burnetii* (14), Jingmen tick virus (12), and *Theileria* (*T.*) *annulata* (11) (Fig. 3).

By the end of 2018, totally 2786, 415, 215, and 129 human cases had been confirmed for infections with *Borrelia* (five species and uncharacterized species), *Anaplasmataceae* (four species), *Babesia* spp. (five species and uncharacterized species), and spotted fever group rickettsiae (six specific species and uncharacterized species), respectively. Additional 216 human cases were infected with other bacteria (three species) including 120 with *Francisella tularesis*, 95 with *Coxiella burnetii*, and one with

*Colpodella* spp. (Fig. 4). In the spotted fever group rickettsiae, *R. heilongjiangensis* and *R. raoultii* were the most widely distributed, covering the west, north, and northeast of China (Fig. 4a; Supplementary Fig. 35; Supplementary Note 2). *R. heilongjiangensis* was also found sporadically in southern China. From 1996 to 2007, human cases infected with *R. heilongjiangensis* were reported in Heilongjiang (1-10) and Jilin provinces (11-12) in the northeast and Hainan Province (11-16) in the south. A few human cases of *R. raoultii* (1-10) were reported in Xinjiang (1-5), Inner Mongolia (6), and Heilongjiang (1-5) provinces in northern China. Henan Province reported five *R. sibirica* spp XY-99 cases, and Anhui reported one *R. sibirica* spp BJ-90 patient. Human cases with uncharacterized *Rickettsia* species were mostly reported in Heilongjiang and Hainan provinces.

As the most commonly recorded agent in the *Anaplasmataceae*, *A. phagocytophilum* was scattered over the whole nation except for the southwest (Fig. 4b; Supplementary Fig. 36; Supplementary Note 2). Most human cases were reported in central and central-east China, primarily in Hubei and Shandong provinces. Cases were also seen in the northeast and the southeast. *E. chaffeensis* had a comparably wide geographic scope, except that it was also detected in the southwest (Yunnan Province), and sporadic human cases were reported in Inner Mongolia, Beijing, Tianjin, Shandong, and Guangdong provinces. *A. capra* was the third most commonly detected agent in humans in the family with a total of about 29 reported case in the northeast, although it was also found in ticks in the central and the west. Another widely distributed agent was *E. canis*, found in the east, the south, and the northwest.

As to *Borrelia burgdorferi* sensu lato complexes in the genus *Borrelia*, which are the etiological agent of Lyme disease, ticks carrying *B. garinii*, *B. afzelii*, and *B. burgdorferi* sensu stricto shared similar distributions across northwestern, northern, northeastern, and southern China, although *B. garinii* was more widely detected in ticks (Fig. 4c; Supplementary Fig. 37; Supplementary Note 2). *B. garinii*, *B. afzelii*, and *B. burgdorferi* sensu stricto are the major causative agents for human Lyme

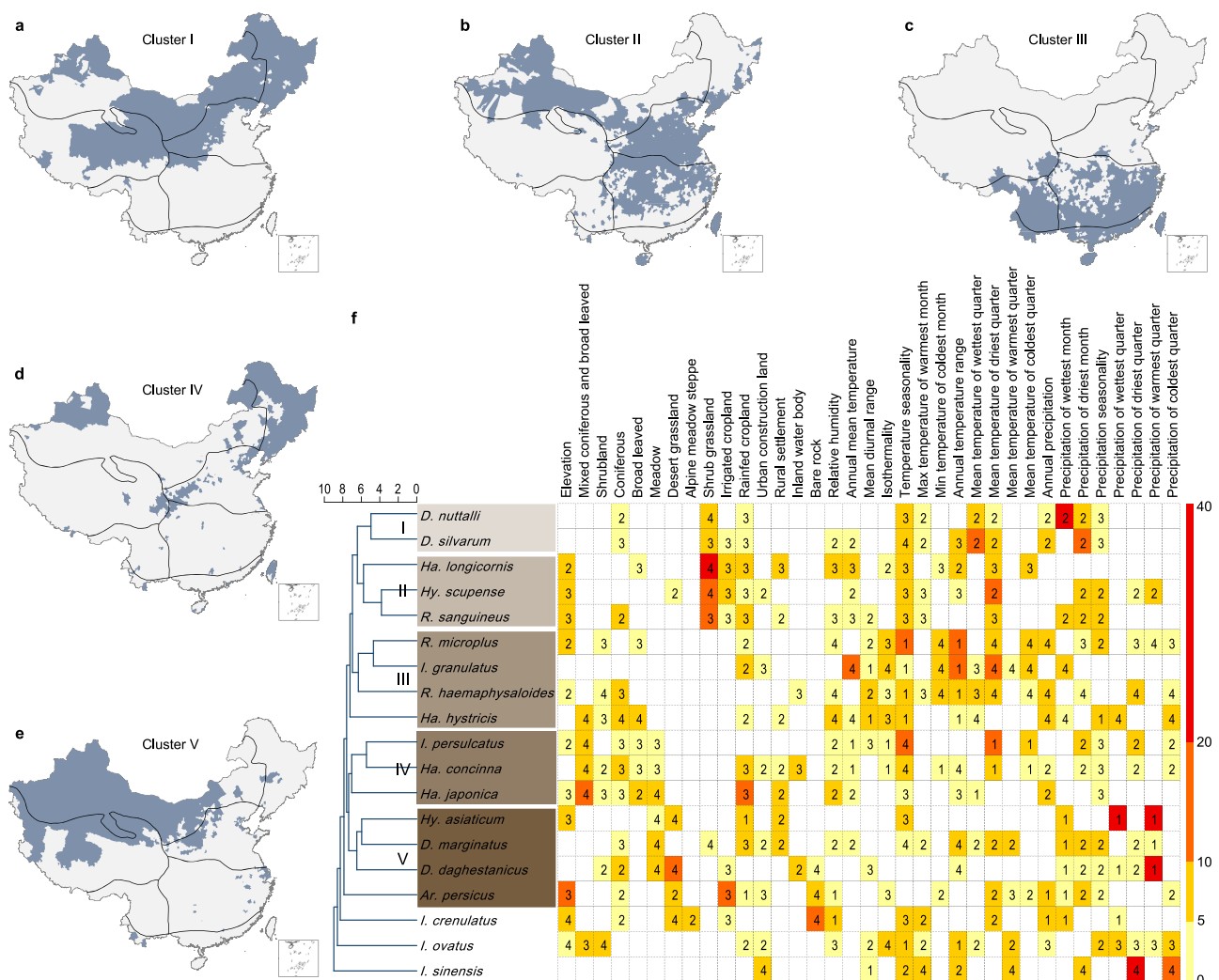

**Fig. 2 Clustering of tick species based on their ecological features and spatial distributions at the county level.** Panels **a–e** indicate the spatial distribution of the five clusters (clusters I–V). The boundaries of the seven biogeographic zones are shown as black solid lines. The dendrogram in panel **f** displays the clusters I–V of tick species. The features used for clustering are three quantities associated with each predictor in the BRT models. Two of the three quantities were displayed in panel **f** to indicate the possible level of ecological suitability: relative contributions (colors in ascending order from yellow to red) and the standardized median value of the predictor (numbers in the heatmap) among counties with tick occurrence (numbers 1–4 indicate the position of this median in reference to the quartiles of this predictor among all counties). Source data are provided as a Source Data file.

disease[20]. The Changbai Mountain area on the border of Heilongjiang and Jilin provinces in northeastern China was a hotspot of human cases where all the four major agents, as well as *B. valaisiana* were found. In addition, cases infected with *B. garinii* were reported in Xinjiang, Inner Mongolia, and Hainan provinces, *B. afzelii* cases were seen in Xinjiang, Chongqing and Shandong provinces, and *B. burgdorferi* sensu stricto infected cases in Shandong and Guangdong. Most Lyme disease pathogens detected in humans were however uncharacterized. *B. miyamotoi* is the causative agent for human relapsing fever, and patients were seen in Heilongjiang and Jilin provinces, northeastern China[21].

The distributions of *Babesia* spp. species were mostly focal, but *Ba. microti* was found in the north, the northeast, the east, the southeast, and the southwest of the country (Fig. 4d; Supplementary Fig. 38; Supplementary Note 2). Human infections with *Ba. microti* occurred in the east near Shanghai and in Yunnan Province of the southwest. *Ba. divergens* was found in human cases in Xinjiang, Gansu, and Shandong provinces, extending from the northwest to the central and to the east of the country, but was detected in ticks only in the northeast. Ticks harboring

*Ba. venatorum* were only found in the northeast, but human infections were reported in both the northeast and the northwest. *Ba. crassa*-like agents parasitized ticks and infected humans in the northeast, mainly in Heilongjiang Province. A human infection with *Ba.* spp. XXB/Hangzhou was recorded in Zhejiang Province, but detection of this agent in ticks has not been reported yet. Human infections with uncharacterized *Babesia* spp. were mostly reported in Inner Mongolia in the north, Zhejiang Province in the east, and Yunnan Province in the southwest.

*T. annulata* was the most widely distributed agent in the *Theileria* genus, followed by *T. sergenti* and *T. luwenshuni*. All three agents were reported in the western, northern, northeastern, and central parts of China, with *T. annulata* also detected in the south (Supplementary Fig. 39; Supplementary Note 2). Other *Theileria* agents, including *T. sinensis*, *T. ovis*, *T. equi*, and *T. uilenbergi*, were only found in Xinjiang in the northwest or Gansu in the central west. Thus far, *Theileria* agents have not been associated with human infection in China.

Of the seven known tick-borne bacteria in China, *C. bumetii*, the causative agent for Q fever, was the most widely distributed, found in either ticks or humans across the country except for the

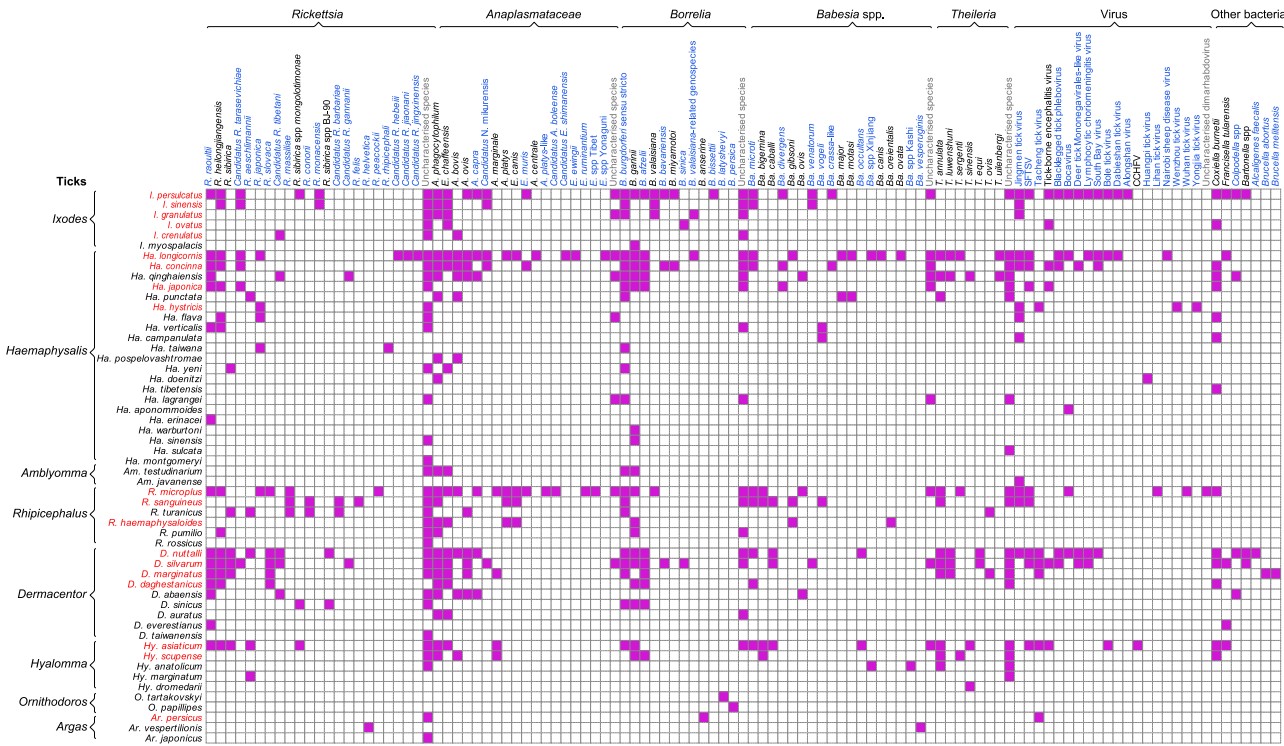

**Fig. 3 The tick species and their corresponding tick-borne agents in China from 1950 to 2018.** The tick-borne agents marked in blue indicate the newly identified agents in the past two decades. Source data are provided as a Source Data file.

central and southern provinces (Fig. 4e; Supplementary Fig. 40; Supplementary Note 2). The distribution of *F. tularensis* was comparable to that of *C. bumetii*, except that it was not detected in Xinjiang and Yunnan provinces. Tibet had a relatively high disease burden for both pathogens, about 46 *C. bumetii* cases and 31 *F. tularensis* cases. Most of the remaining patients were reported in Shandong Province for *F. tularensis* and in Yunnan Province for *C. bumetii*. *Bartonella* spp. and *Alcaligenes faecalis* were found in ticks in northeastern China, particularly around the Daxing'an Mountains. *Colpodella* spp.-carrying ticks were only found in the northern part of Qinghai–Tibet Plateau, but a human case was reported in Heilongjiang Province in 2013[22]. *Brucella melitensis* and *Brucella abortus* were found in *D. marginatus* ticks in northwestern Xinjiang. Brucellosis is a common zoonotic disease in China, predominantly caused by *Brucella melitensis*[23]. However, as the transmission of brucellosis by ticks is rare, we do not show human cases on the map.

Altogether 19 tick-borne viruses have been identified in China, six of which were associated with human patients (Fig. 4f and Fig. 5; Supplementary Fig. 41; Supplementary Note 2). Among them, SFTSV and TBE virus (TBEV) were responsible for the enormous disease burden unparalleled by any other tick-borne pathogens (Fig. 5). Hundreds of human cases infected by Crimean–Congo hemorrhagic fever virus (CCHFV) were reported in Xinjiang, with a few reported in Yunnan Province (Fig. 4f). Jingmen tick virus was first found in 2010 and was distributed in central, eastern, and northeastern China (Supplementary Fig. 41), with 12 human cases reported in Heilongjiang Province (Fig. 4f). Another recently discovered agent, Alongshan virus (ALSV), was detected in ticks in the northern tips of Inner Mongolia and Heilongjiang, as well as in human cases across the two provinces.

**Risk mapping and risk factors for pathogens associated with major TBDs.** The majority of human SFTS cases during 2010–2018 were diagnosed in Liaoning Province in the northeast,

Shandong, Jiangsu, and Zhejiang provinces on the east coast, and Henan, Hubei and Anhui provinces in central China (Fig. 5a). The etiological virus, SFTSV, was detected primarily in *Ha. longicornis*, but also in *D. nuttalli* in northern Xinjiang. The model-predicted risk areas resembled the current reporting regions (Fig. 5b). Approximately 251.5 million people reside in high-risk areas where the model-predicted probability of SFTSV presence exceeds 50%. Temperature seasonality, mean temperatures during the wettest quarter, elevation, annual temperature range, closed woodland, mean temperatures during the driest quarter[24] were the leading risk determinants for the presence of SFTSV with RC >7% (Table 2). SFTSV ecologically prefers regions at low to moderate elevations (<1000 m) and with strong seasonality and wide annual variation range in temperature, a low mean temperature in the winter (driest quarter), and a high mean temperature and precipitation in the summer (wettest quarter or month) (Supplementary Fig. 42).

Human cases of TBEV primarily clustered in northeastern China, coinciding with the model predicted high-risk areas (Fig. 5c). The northwestern region where the virus was found only in ticks was classified as having mild to moderate risks (Fig. 5d). In total, about 94.5 million residents live in the high-risk areas. Temperature seasonality was by far the most influential predictor with RC = 54.0% (Table 2), which was also a leading predictor for the presence of *I. persulcatus*, a major carrier of TBEV (Supplementary Table 1). Additional important contributors included mean temperature in the driest quarter, elevation, and closed woodland (RC > 7%). High risks of TBEV were flagged by low to medium elevations (<1000 m), strong seasonality in both temperature and precipitation, and low temperature in both winter (driest quarter) and summer (warmest month) (Supplementary Fig. 43).

## Discussion
We assembled the most comprehensive, if not all, records of tick species and tick-borne pathogens in China that cover a time span

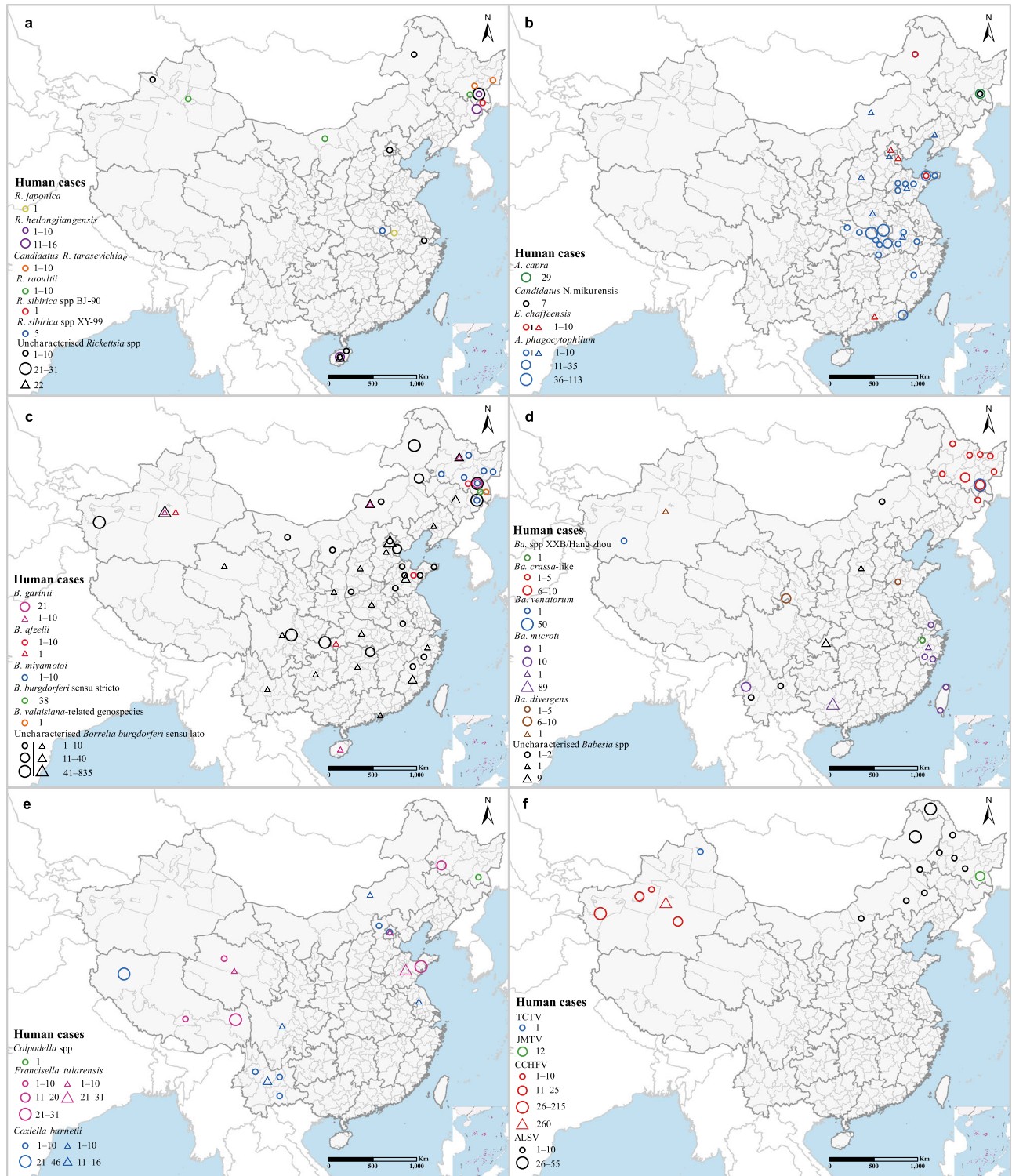

**Fig. 4 The distributions of human cases by species of tick-borne agents in China during 1950–2018.** Human cases are positioned at the center of either province (triangles) or prefectures/counties (circles) depending on data availability. **a** spotted fever group rickettsiae; **b** *Anaplasmataceae*; **c** *Borrelia*; **d** *Babesia* spp.; **e** bacteria; **f** viruses. Human cases of SFTSV and TBEV are not shown as they are described in other figures. Another five tick-borne viruses, including Huangpi tick virus, Lihan tick virus, Wenzhou tick virus, Wuhan tick virus, and Yongjia tick virus, were not displayed in the map due to lack of location information. Source data are provided as a Source Data file.

of 70 years. We reported detection locations of some tick species not covered by existing compiled data sets[16], e.g., *H. longicornis* in Xinjiang and *I. persulcatus* in Jiangsu (Supplementary Figs. 2, 3). Our cross tabulation of tick species and associated pathogens is also more complete than previous studies, e.g., listing ten known

pathogens detected in *H. japonica* and nine tick species harboring SFTSV, compared to only two and one, respectively, in a previous review[17]. Using a robust machine-learning algorithm, we found that the geographic scopes of major tick species (particularly *Ha. longicornis*, *I. sinensis*, *R. microplus*, *R. sanguineus*, and

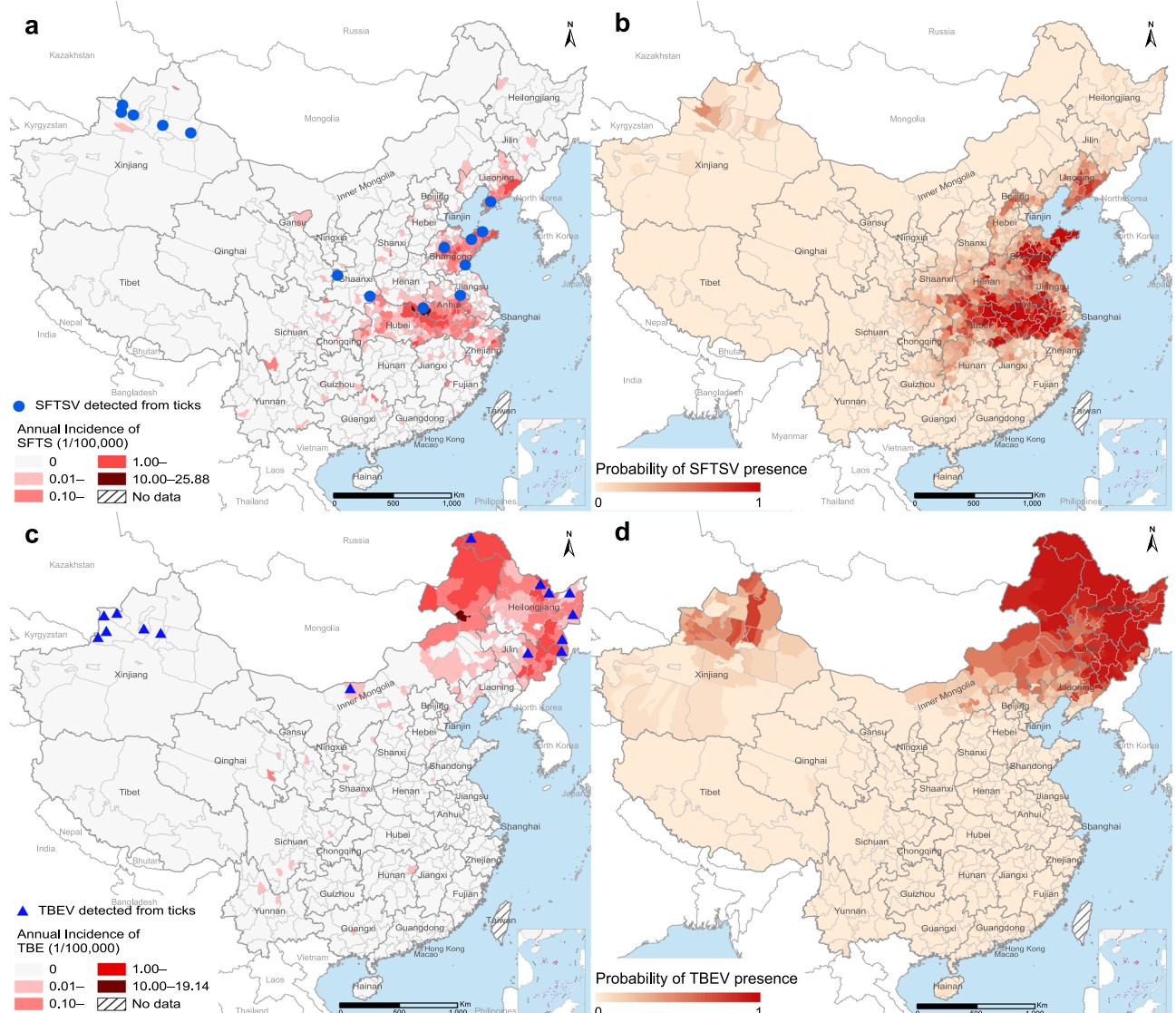

**Fig. 5 The reported and model-predicted distributions of SFTSV and TBEV at the county level in China. a** Reported annual incidence rate of human SFTS and locations of SFTSV detected from ticks. **b** Spatial distribution of model-predicted probabilities of SFTSV presence. **c** Reported annual incidence rate of human TBE and locations of TBEV detected from ticks. **d** Spatial distribution of model-predicted probabilities of TBEV presence. Source data are provided as a Source Data file.

*R. haemaphysaloides*) could be up to 5-fold as large as what has been observed, likely due to limited field investigations or incomplete sampling (Table 1). However, it is also possible that our models were underspecified and thus overestimated the scopes.

The ecological niches for ticks are complex, and the key predictors differ even within the same genus. For example, *Ha. concinna*, *Ha. japonica* and *Ha. hystricis* prefer places covered by coniferous or broad-leaved trees, whereas *Ha. longicornis* thrives in shrub grasslands. It is therefore meaningful to group tick species by their ecological characteristics, in addition to their genera, to better understand the overall risk of tick exposure at any given place. We found five clusters of tick species that share comparable ecological niches and geographic distributions. Such clustering offers additional information for risk assessment and field investigation. For instance, despite the low detection and low risk of *Ha. longicornis* in northwestern China (Supplementary Figs. 3, 31), it should be targeted for survey in this region where both *R. sanguineus* (Supplementary Figs. 5, 32) and *Hy. scupense* (Supplementary Figs. 7, 34), which are in the same ecological

cluster as *Ha. longicornis*, have high prevalence of field detection and model-predicted risks.

*Ha. longicornis* is by far the most widely distributed and influential tick species, exposing over 40% of the nation's population in 1140 counties of eastern and northeastern China. The underlying implication for public health is enormous, as *Ha. longicornis* harbors 44 tick-borne pathogens and is a competent vector for SFTSV that was associated with a case fatality ratio of 12–50%[25]. Native to East Asia, *Ha. longicornis* was thought to be imported from Japan to Australia and New Zealand in the 19th century[7]. The recent emergence of *Ha. longicornis* in eight states of the U.S. suggests a global spread and flags the need for close monitoring of the ticks and related pathogens in this regions[8,26].

The genus of *Ixodes*, mainly comprising *I. persulcatus*, *I. sinensis* and *I. granulatus*, also imposes serious threats to public health. *I. persulcatus* is a major carrier for both TBEV and *B. burgdorferi* sensu lato (*Borrelia*), the latter also being carried and transmitted by two other Ixodes species[27,28]. The three tick species have distinct ecological habitats that jointly cover many

**Table 2 BRT-model-estimated mean (standard deviation) relative contributions of major ecoclimatic and environmental factors (RC ≥ 4%) to the spatial distributions of SFTSV and TBEV.**

| Category | Variable | SFTSV | TBEV |
|---|---|---|---|
| Ecoclimatic | Isothermality | 6.07 (1.28) | |
| | Temp. seasonality | 16.89 (2.84) | 53.97 (7.82) |
| | Annual temp. range | 9.46 (1.34) | |
| | Mean temp. wettest quarter | 10.07 (1.32) | |
| | Mean temp. driest quarter | 7.11 (1.12) | 27.43 (5.24) |
| | Mean temp. coldest quarter | | 5.24 (1.93) |
| | Precip. wettest month | 5.84 (1.09) | |
| | Precip. driest quarter | 5.35 (1.13) | |
| Environmental | Elevation | 9.97 (1.89) | 7.65 (1.60) |
| | Closed woodland | 9.03 (1.20) | 7.24 (1.39) |
| | Shrubland | 6.01 (0.79) | |
| AUC | Train | 0.977 (0.970-0.985) | 0.978 (0.969-0.987) |
| | Test | 0.897 (0.874-0.920) | 0.894 (0.862-0.926) |
| Partial AUC | Train | 1.69 | 1.81 |
| Ratio | Test | 1.50 | 1.64 |

Mean AUCs (95% percentiles) and partial area AUC ratio (calculated at tolerance level of 0.2) are given.

densely populated areas, particularly in central, southern, and eastern China. *I. sinensis* alone exposes 363 million people in 630 counties, only second to *Ha. longicornis*.

Among the ecoclimatic factors, the most influential for tick ecology are temperature seasonality and the mean temperature in the driest quarter. The driest quarter as well as the coldest quarter overlap with the winter season in most areas. Consequently, the survivability in the winter season is a key to the ecology of ticks. While grasslands, croplands, and woods constitute the natural habitat for ticks, our analyses indicate the nontrivial role of human settlements. For example, both *Ha. longicornis* and *Ha. japonica* had a positive association with the coverage of rural settlement, indicating their tendency of parasitizing domestic animals[29,30]. In contrast, both *Hy. asiaticum* and *D. marginatus* seem to prefer grasslands and croplands with fewer rural settlements (Supplementary Tables 4, 5). The potential preference of *I. sinensis* for a high coverage of urban constructions warrant further investigation given the fast urbanization in China (Supplementary Table 1).

Our findings on ecological drivers for SFTSV and TBEV, two notifiable tick-borne pathogens in China, bear some similarity with our previous ecological studies on SFTS and TBE diseases in human[27,31,32]. For example, elevation, close-canopy woodland, shrubland, and precipitation in the driest quarter were also drivers for SFTS[32], and elevation and close-canopy woodland were drivers for TBE[27]. However, the current work identified more climatic conditions that are important for the ecology of the pathogens, e.g., temperature seasonality and mean temperature in the driest quarter for both SFTSV and TBEV, and even more climatic drivers for SFTSV (Table 2). In addition, the current study identified a wider geographic area of 393 counties (522,000 km²) with a larger population (251.5 million people) at risk of potential SFTSV occurrence, compared to 222 counties (324,000 km²) covering 142.2 million people in one previous study[31], and 384 counties (579,000 km²) covering 226.5 million people in another[32]. In particular, we found that the Xinjiang Uygur Autonomous Region in northwestern China and areas in central and northern China are subject to an increasing yet largely neglected risk of SFTSV infection[31,32]. The model-predicted area and population at risk of TBEV occurrence are also larger than those shown in a previous study (278 vs. 214 counties, 1,609,000 vs. 1,430,000 km², and 94.5 vs. 68.4 million persons)[27]. The additional at-risk areas regarding TBEV found by the current

study are mainly distributed in the south of northeastern China which is densely populated[27]. The differences in the results may come from differences in data and methodology, e.g., our TBE data are more up to date, and our choice of control sites for ecological modeling weighed in existence probability of the competent ticks. Most importantly, the current study focuses on the ecology of the pathogens rather than human diseases, fundamentally different from the previous studies.

Although the risk of SFTSV occurrence is low to moderate in the northwestern area (northern Xinjiang), more active surveillance of both human cases and SFTSV-carrying ticks are recommended for several reasons: (1) SFTS was reported in a tourist to this area recently[33], (2) SFTSV was detected from both *D. nuttalli* and *Hy. asiaticum*, two prevalent tick species in the area[33,34], and (3) this area lies in the heart of the Central Asia Flyway where ticks may exchange among distant regions via migrant birds[35]. For TBEV, we also recommend surveillance and prevention in northern Xinjiang, although no human cases have been reported there. Our models were built on data only from China and need to be further validated using data from other countries, e.g., South Korea and Japan where SFTS is also endemic.

As Lyme disease is not a notifiable disease in China, many human infections with *B. burgdorferi* sensu lato can only be located to provinces rather than counties. Some human cases were reported in the central north and central east of China, e.g., Shandong and Henan provinces, where model-predicted risks for the Ixodes are low (Supplementary Figs. 3, 38). On the other hand, these areas have moderate to high risks for *R. sanguineus*, *R. microplus* and *Hy. scupense*, which are also known to harbor *Borrelia* (Fig. 3, Supplementary Figs. 5, 7)[36,37]. More active field surveillance of ticks capable of transmitting Lyme disease and related pathogens is recommended for these regions.

Our study is subjected to a few limitations. Firstly, the survey locations of ticks were unlikely sampled randomly. Consequently, a certain level of bias could exist in the modeling results. Most likely, biased sampling might have biased our analyses towards the null, i.e., contributions of some important factors might have been underestimated if their distributions had influenced the sampling of survey sites. However, the surveyed 1134 counties (39% of all counties in China) cover all biogeographical zones of China (Supplementary Fig. 1), implying a low possibility of geographic bias. Secondly, the land cover data used for ecological

modeling of ticks were collected in 2005, which may not be appropriate given the rather rapid landscape change in China in the recent decade. On the other hand, while some tick detection records were obtained post 2010, the establishment of their habitat likely had evolved much longer. Finally, while the AUC values are high for all the models we fitted, AUC does not necessarily reflect the goodness of fit and could be misleading when the absence data are associated with high uncertainty[38]. Such uncertainty exists as most surveys are cross sectional.

In conclusion, our study found it necessary to expand current field survey efforts for ticks, especially those harboring pathogens implicative of imminent threats to public health. Meanwhile, it is wise to strengthen surveillance for TBDs by increasing diagnosis and treatment capacities in areas where human cases have emerged or the model-predicted risk level is high. As urbanization and the Grain-to-Green Program (restoring forests from croplands) are ongoing in parallel in China, the dynamics of ticks and tick-borne pathogens as well as diseases will be complex and need close monitoring.

## Methods

**Data on ticks and tick-borne pathogens**. We assembled a comprehensive database of ticks and tick-borne pathogens (the database is available upon request) from a variety of sources, including (1) literature reporting the occurrence of 124 ticks and 103 tick-associated agents in China, published between January, 1950 and December, 2018 (Supplementary Fig. 44, Supplementary Table 6), (2) historical data (before 1990) on presence records of ticks across China that are not formally published but available in the Medical Entomology Gallery (MEG) and unpublished data on the prevalence of ticks or tick-borne pathogens from entomological surveys conducted by our institute in mainland China from 1990 to 2018[39], and (3) newly conducted field surveys of tick species across the country[14,40-42]. All the entomological surveys in literature and conducted by our institute were cross-sectional studies. For the literature review, five main electronic databases (PubMed and ISI Web of Science, China WanFang database, China National Knowledge Infrastructure, and Chinese Scientific Journal Database) were searched for studies published between January, 1950 and December, 2018, using the following keywords: ("Tick" or "Ticks") and "China". We also checked the references in retrieved articles to reach more relevant articles. Each article was carefully reviewed by two team members independently to collect the following information using a standard form: study date, study location, spatial resolution, tick species identified, laboratory methods, and detection results for tick-borne pathogens. Any disagreement between the two staff members was resolved by discussion and consensus among the reviewers and other co-authors. Only studies with clearly identifiable results, i.e., presence or absence, time and location of tick species or tick-borne pathogens were included in our database (Supplementary Table 6). If a tick species or tick-borne pathogen was reported more than once in the same county (e.g., through seasonal collections or by different study groups or in more than one townships within a county) during the study period, it was counted only once in our analyses. The tick-borne pathogens included in our database were detected from ticks by any of the following laboratory tests: real-time polymerase chain reaction (RT-PCR), PCR, isolation or culture, or smear microscope. If more than one pathogen were found in the same study or isolated from the same tick, a record was created for each pathogen in our database. For articles containing ambiguous data, the original authors were contacted for clarification; if the ambiguity was not clarified, the data in question were excluded from our database. These literature-extracted data and data from other sources were integrated to form one database at the county level for final analyses.

**Data on socioenvironmental and ecoclimatic factors**. We collected a variety of environmental and climatic variables that are commonly used in ecological studies on the spatial distribution of tick species and tick-borne pathogens[15,31,43,44]. The choice of variables is mainly based on empirical ecological evidence in the literature and their spatial variability. In addition, we focus on ecological variables that are potentially shared by multiple species so that the results can be compared across species. For example, although our previous study has shown that the land cover of tea farms contributes to the distribution of SFTS patients and hence is a potential predictor for the presence of *Ha. longicornis*, we did not include this variable because there is no evidence for the association of tea farms with most other tick species.

The 38 years (1981 to 2018) of climatic data were collected from 2006 weather surveillance stations in mainland China, covering 71.3% of 1134 surveyed counties (http://cdc.nmic.cn/home.do). The climatic data include average monthly meteorological variables such as temperature, maximum temperature, minimum temperature, relative humidity, and rainfall during the 38 years. For the 877 counties (326 with tick presence) without meteorological stations, the mean values

of the nearest five surveillance stations were used as a proxy for their meteorological variables. From these longitudinal meteorological variables, 19 cross-sectional ecoclimatic variables (BIO01-19, also called bioclimatic variables recommended by the U.S. Geological Survey) were created and their yearly averages were used as predictors in our risk models[24,45]. These ecoclimatic variables better capture the seasonal trends of different species related to their physiological constraints than traditional meteorological variables and have been widely used in ecological studies[24].

China updated its land cover data every 5-10 years since 1995. Data sets from different years may not be directly comparable or combinable as land cover categories and classification criteria often changed. Raster-type land cover data of China in the year 2005 and 2015 with a resolution of one square kilometer were obtained from the National Earth System Science Data Sharing Infrastructure (http://www.geodata.cn). Our tick records span over the past five decades. While many tick surveys were conducted after 2010, the establishment of the ticks' habitats likely has evolved much longer. Therefore, we used the 2005 land cover data to model the distributions of tick species. We used the more recent 2015 land cover data for modeling the distributions of SFTSV and TBEV because a large portion of the data was generated over the recent decade, in particular for SFTSV (after 2010). Elevation data were obtained from the Shuttle Radar Topography Mission (SRTM) archives (http://www.srtm.csi.cigar.org).

Given that many ticks feed on domestic animals and the majority of patients affected by tick-borne diseases in China were rural residents, we extracted demographic data in the form of the proportion and density of rural population from the 2010 census data obtained from the National Bureau of Statistics of China (http://www.stats.gov.cn/). We strictly limited the number of demographic and socioeconomic variables to (1) avoid model overfitting as the survey data of tick species are rather limited, and (2) focus on variables with a direct rather than indirect link to the ecology of tick species and tick-borne pathogens so that the results are more generalizable to outside China.

In total, 45 socioenvironmental and ecoclimatic variables at the county level were extracted from these data using the ArcGIS 10.0 software (ESRI Inc., Redlands, CA, USA) (Supplementary Table 7). Data cleaning and reorganization with regard to these variables, such as the calculation of the proxy climatic variables for counties not covered by meteorological stations, were performed in the statistical software R (ver. 3.6.0).

**Data on clinical cases of tick-borne pathogens**. Clinically diagnosed or laboratory-confirmed TBE and SFTS cases at clinics and hospitals are reported, as mandated by the Ministry of Health, to the Chinese Information System for Diseases Control and Prevention. The county-level data of human cases of TBE and SFTS during 2005-2018 were collected from the Chinese Scientific Data Center for Public Health (http://www.phsciencedata.cn) and were used in the ecological models for tick-borne pathogens.

**Spatial mapping**. Recorded occurrences of ticks, tick-borne agents, and human cases were geo-referenced at the county level when data permit or at the prefecture or province level otherwise. In total, 78.8% of tick species, 63.3% of tick-borne pathogens (100% for SFTSV and TBEV), and 60.6% of TBDs were geo-referenced at the county level. All maps were produced using the ArcGIS 10.0 software.

**Ecological modeling**. For each of the 19 major tick species, a case-control study design was used to build predictive machine-learning models at the county level. Records that could not be geo-referenced at the county level were excluded. Briefly, for each given tick species, counties with at least one record of occurrence were considered as "cases", and those surveyed but lacking any evidence of occurrence were considered as "controls"[46]. The numbers of "cases" and "controls" for each tick species were listed in Supplementary Data 1. The remaining counties where tick surveys have not been conducted or have not yielded conclusive findings were excluded from model building but were included for risk mapping. For example, among a total of 1134 counties where ticks were surveyed and tick species were determined, 382 counties recorded occurrence of *Dermacentor nuttalli*, and were thus considered "cases", and the other 752 counties were considered "control" sites, for modeling the distribution of *Dermacentor nuttalli*. A Boosted Regression Trees (BRT) model at the county level was fitted to the training set to assess the contributions of ecoclimatic and socio-environmental predictors to the geographic distribution of the given tick species. The BRT model is a popular approach to ecological studies and has been widely used for risk mapping of infectious diseases such as avian influenza, rabies, and helminth[47-50]. The BRT model couples the advantages of two algorithms, regression trees and machine learning techniques, and allows nonlinear relationships between outcomes and covariates and multi-collinearity among covariates[51]. For each BRT model, 43 variables including 24 environmental and 19 ecoclimatic factors were used as potential predictors (Supplementary Table 7) for the presence and absence of the tick species in each county. The fitted model was used to project risk levels in counties without tick surveys[52,53]. To counterbalance potential sampling bias of survey counties, we built a logistic regression model for the selection of tick survey counties with all ecoclimatic and socio-environmental variables as predictors (Supplementary Table 7). The response of this model was one for all tick-surveyed counties and zero for

unsurveyed counties. The predictors were chosen using a backward procedure at the significance level of 0.05 (Supplementary Table 7). The reciprocals of predicted sampling probabilities of all surveyed counties were first rescaled to have a mean of one and then used as weights in the BRT models for the 19 major tick species[54–56].

A tree complexity of five, a learning rate of 0.005 and a bagging fraction of 75% were used for the primary analysis based on their satisfactory performance in our previous research[57,58]. Bagging is a procedure that resamples data points to fit sequential trees so to improve predictive performance. A 10-fold cross validation was used to identify the optimal number of trees using the gbm.step function in the R *dismo* package. The output of a BRT model consists of both predicted probabilities of occurrence and relative contributions (or influences) of predictors. The relative contribution is calculated based on how many times a predictor is chosen for splitting and how much each split improves the objective function, averaging over all trees. These relative contributions of all predictors are standardized so that they sum to one[51]. A two-stage bootstrapping procedure was employed to provide a more robust and parsimonious estimation of model parameters. In each stage, the following split-and-fit step was repeated for a certain number of times. A training set with 75% of data points was randomly selected by bootstrapping without replacement, and the remaining 25% served as a test set. A BRT model was built using the training set, and then applied to the test set for validation if needed. In the first stage, the split-and-fitting step was repeated for ten times to screen important predictors. Validation of the training model using the test set was not performed in this stage. Predictors that had a relative contribution <2% for all bootstrap training sets were excluded from the next stage. In the second stage, the split-and-fitting step was repeated for 100 times using the remaining predictors. As no variable selection was performed in this stage, all 100 models had the same predictors but yielded different contribution estimates. The relative contributions of the predictors were averaged over the 100 BRT models to represent their final relative contributions. The ROC curves and areas under the curve (AUC) based on the test sets were also averaged to represent the final predictive performance. The standard deviations and 95% percentiles of the relative contributions and AUCs across the 100 models were used to quantify the uncertainty in the estimation. Considering that there could be false negative and false positive counties in the observed data, we also calculated partial area AUC with a tolerance level of 0.2 for omission error[59]. For partial area AUC, the horizontal axis is the total rate of positives rather than false positives. we presented the ratio of the partial AUC to the area under the random selection line (diagonal line) as suggested by Peterson et al.[59]. Finally, the predicted probabilities were averaged over the 100 models to represent the final estimates of the county-specific probabilities of presence, which were mapped for the 19 main tick species[47,48,52,53]. BRT Modeling was conducted using the R packages dismo and gbm, and predictive performance was assessed using ROCR and pROC in the R v3.6.0 environment (https://www.r-project.org). We also performed a sensitivity analysis using a learning rate of 0.01 for selected tick species but found no substantial difference in the contribution estimates. Due to both the data size (45 predictors) and the number of models runs ([19 ticks + 2 pathogens] × 100), we cannot afford a full cross-validation optimization for all model configuration parameters.

To determine model-predicted high-risk counties for each tick species, we chose a cut-off value that maximizes sensitivity + specificity along the ROC curve for each final BRT model[60,61]. Counties with predicted probabilities above the cut-off value for a given model were considered as having a high risk of harboring the corresponding tick species. For each tick species, the number, area, and population size of model-predicted high-risk counties were compared to the quantities of counties with observed occurrence (Table 1).

**Clustering ticks with similar ecological niches and their spatial distribution**.
To explore similarity in ecological niches among the 19 predominant tick species, a hierarchical cluster analysis based on the weighted-average linkage method was performed[62]. Features used for clustering were formed as the following. We first excluded predictors that are not influential (excluded from final models) for all 19 ticks. For each tick species, three quantities associated with each remaining ecological predictor were calculated as features for clustering. One is the average relative contribution of this predictor in the final 100 BRT models. If the predictor was not included in the final models for this tick species, its relative contribution was set to zero. The second quantity is a measure for the difference in this predictor between case counties (positive for the given tick species) and all counties. We first calculated the median value of this predictor among all case counties and quartile intervals of the predictor among all counties in the nation. We then assigned one of the numbers 1-4 according to which quartile interval the median lies in, e.g., assign 1 (4) if the median lies in the lowest (highest) quartile. The third quantity is the linear correlation between the predictor and model-predicted presence probabilities of the given tick species among all counties (averaged over the 100 models). These three quantities of all ecological predictors jointly serve as features for clustering. A dendrogram was created to demonstrate the clustering pattern of these 19 tick species, together with a thematic matrix illustrating the features (Fig. 2). This matrix has tick species as rows and predictors as columns. The color of each cell in the matrix shows the average relative contribution and the number shows the quartile (1-4 for 1st-4th quartiles) location of the median of cases. To map geographic distributions of the identified clusters of tick species at the county level, we define the presence of each cluster as the presence of any tick species in that cluster.

**Population at risk for emerging tick-borne pathogens**. BRT models were also used to evaluate the risk and risk drivers for the presence of the etiological pathogens for SFTS and TBE, the two most commonly reported tick-borne diseases (TBD) with mandated surveillance in China. For each pathogen, the corresponding model considers the same 45 potential environmental and ecoclimatic predictors used for modeling tick species. All counties where the pathogen was detected in ticks according to literature or human cases of the associated TBD were reported by surveillance were regarded as "cases". For a county to be assigned as a "control", the following conditions must be satisfied: (1) No human cases of the associated TBD and any other evidence for the presence of the pathogen were reported by surveillance or literature; (2) The primary tick vector (*Ha. longicornis* for SFTSV and *I. persulcatus* for TBEV) was surveyed but not found; or the primary tick vector was not surveyed but the probability of the existence of this tick species, as predicted by the corresponding BRT model, is smaller than the cutoff that yields the best predictive performance of the model (represented by the Youden's index).

We made these stringent criteria for controls because it is much harder to exclude the possibility of the existence of any pathogen. Note that a county where the tick vector was detected but tested negative for the pathogen does not qualify this county as a control, because even in "case" counties, detection rates of a pathogen among field-collected ticks are usually low. Our choice of control counties may result in a certain level of overestimation of high-risk counties, yet overestimation is preferred to underestimation from the perspective of disease prevention. A two-stage bootstrap procedure was also used to generate 100 BRT models for each TBD, based on which the average relative contributions were calculated and the average predicted probabilities of TBD were mapped.

**Reporting summary**. Further information on research design is available in the Nature Research Reporting Summary linked to this article.

## Data availability
Model results generated as part of this study and the raw data that support the findings of this study are available within the paper and its supplementary information files. Source data are provided with the paper.

## Code availability
The R code used to implement the BRT models is provided in Supplementary Code 1.

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

## Acknowledgements

We thank Dr. Wuchun Cao, Dr. Tao Jiang, and Dr. Yi Sun from the Beijing Institute of Microbiology and Epidemiology for participating in the discussion of the analysis strategy. This work was supported by the Special Program for Prevention and Control of Infectious Diseases in China (2018ZX10201001, 2018ZX10101003, 2018ZX10713002, and 2017ZX10303401), the National Science Foundation for Distinguished Young Scholars of China (81825019). Y.Y. was supported by the U.S. National Institutes of Health grant R56 AI148284.

## Author contributions

L.-Q.F., W.L., and Y.Y. designed the study. G.-P.Z., Y.-X.W., Z.-W.F., Y.J., W.-H.Z., and S.-X.Z. performed the literature review, data collection, and integration. G.-P.Z., Y.-X.W., S.-X.Z., X.-L.L., and M.-J.L. conducted the analyses under the supervision of L.-Q.F., W.L., Y.Y.. H.L. and S.L. helped with the analyses. Y.Y., W.L., and L.-Q.F. wrote the draft of the manuscript. All authors contributed to and approved the final version of the manuscript.

## Competing interests

The authors declare no competing interests.
