## [Peer Review File · Nature Communications]

Reviewers' Comments:

Reviewer #1:

Remarks to the Author:

This manuscript reports on the spatial distributions of ticks and tick-borne pathogens in China based on a review study of relevant literature and public health data. China is a country prone to high risks of tick-borne diseases, as ticks distribute widely over the country and green areas suitable for tick survival are significantly increased over the past few years. As the topic is important, an increasing number of review articles on the same topic have been published in the past few years, overlapping most of the aspects this manuscript covers. While this manuscript failed to address, challenge or even compare with some of the important existing review studies, this questions the level of novelty and originality of this work, at least from its current stand, to be justified as a potential publication in highly-ranked journals like Nature Communications.

My primary concerns are:

1. This comprehensive work is built on a review of tick and tick-borne pathogen occurrence studies conducted in China (1950-2018). It summarizes the distribution of 124 tick species (at the county-level), and the associated pathogens carried by 56 tick species. In general, the manuscript misses providing a current understanding of ticks' biogeography and taxonomy in China as provided in several recent relevant review studies and comparing itself with them to update such an understanding. This makes it look as if the authors lacked the scientific ground they stood on or that they were avoiding these studies because of the overlapping outcomes. These studies are:

(a) Zhang et al. (2018, Archives of Insect Biochemistry and Physiology, DOI: 10.1002/arch.21544) conducted a mini-review on the distribution of 125 tick species as well as their corrected names in China at the province level;

(b) Zhang et al. (2019, Scientific Data, DOI: 10.1038/s41597-019-0115-5) reviewed the location records of tick in China reported 1960-2017 and published point-based, open access spatial data on the distribution of 123 tick species ;

(c) Yu et al. (2015, Parasites & Vectors DOI: 10.1186/s13071-014-0628-x) reviewed articles and provided a table of a range of potential disease causative agents in 52 tick species found in China. Thus, the existing evidence has already built a sound, comprehensive and reproducible outcome regarding the current distribution of ticks and the pathogens in China. The originality this manuscript may add on them should be based on an insightful analysis on the differences from or the corrections to the existing knowledge. My recommendation is to compare with these studies, and some others, and update the understanding of the distribution and taxonomy of ticks and the pathogens they harbor in China.

2. One of the main results of this manuscript is on the mapping and prediction of the potential distribution of SFTSV and TBE in China using the boosted regression tree (BRT) model. These parts of the work were based on previous studies published by the same authors or authors from the same institution. Fang et al. (2015, The Lancet Infectious Disease, DOI: 10.1016/S1473-3099(15)00177-2), reviewed existing and emerging tick-borne infections in mainland China, with the distribution of SFTSV being one of the foci of the paper. Sun et al. (2017, Ticks and Tick-borne Diseases, DOI: 10.1016/j.ttbdis.2017.04.009) reviewed and accessed the evidence on the distribution of ticks and TBE incidence across China, and mapped the potential distribution of TBE using the same BRT approach. Sun's results on the predicted TBE map looks very similar to the one represented here in the manuscript. As a comparison with these previous works was not conducted either, it seems that only minor updates have been made compared with these studies published five or six years ago. My recommendation is to explicitly explain/discuss the new/different findings of this work.

3. The design of the BRT modeling approach may need a better explanation and justification.

(a) Rationales behind the selection of predictors and data processing steps are missing (Appendix pp.6-7; Supplementary Table 9). It reads that "climatic data were collected from 2006 surveillance

stations in mainland China, covering 68.7% counties". Then, how many counties with tick presence did not have surveillance stations? Besides, why to include such a set of the 19 eco-climatic variables? Why to only include climatic data between 1981-2010, presumably the post-2010 data records on tick and tick-borne pathogens were more abundant (Zhang et al., 2019)? Why to use the land cover dataset of 2005, provided that there has been a continuous rapid land cover change in China thereafter? Why to include rural population (proportion and density) as predictors when modeling the distribution of ticks and pathogens?

(b) While conducting the case-control design to select data points for BRT modeling, it reads that "those with surveys conducted but without detection of the tick species were considered as 'control' sites" (Appendix pp.7-8). Since the sample size matters, it would be better to provide at least the number of 'control' sites when modeling each of the 19 tick species. This could enhance readers' confidence in accepting these predictions.

(c) Finally, in the abstract, it reads that "major tick species are demonstrated to have much more extensive suitable habitats than what have been observed". This is somewhat not surprising because every species distribution model will produce predictions like that.

Overall, I think the manuscript has aimed too high to compile all the relevant knowledge on ticks and tick-borne disease in China, making it losses focuses and lacks clear/key take-home messages. I do appreciate the authors' effort on conducting such an organized and comprehensive study like this. However, I would regard this work more suitable for specialized journals, such as International Journal for Parasitology, Parasites & Vectors, and Ticks & Tick-borne Diseases.

Reviewer #2:

Remarks to the Author:

I this study Zhao et al. has collected comprehensive data on tick species in China abundance and described their ecological niche using boosting regression trees. Along with developing models for the nineteen most common tick species found in China they also present a model describing the spatial risk of tick-borne pathogens specifically Tick-borne encephalitis virus (TBEV) and Severe Fever with thrombocytopenia syndrome virus (SFTSV). The detailed data presented here in the manuscript is one of the important strengths of the study. Keeping in mind the importance of ticks and tick-borne infections for public health the scope of the manuscript is relevant. Authors do a great job in describing the selection of predictors, case-control experimental set and use of randomly selected 75% data as the training set to test on the remaining 25% of the data. However, I have some concerns related to the validation of models presented in the study, especially hyper-tuning of boosted regression trees and validation through the cross validation-holdout method. Details of which are lacking the manuscript.

Major concerns related to analysis:

1. Cross-validation and optimum model parameters: It is not clear on how boosted regression tree (BRT) models were developed and what exactly was the outcome variable of interest and how the corresponding BRT model was developed. For example, if the model is used to predict presence and absence of (binary 1,0) of tick species or abundance of tick species is unclear? The authors do not present the details of the BRT model parameters used for all the models developed here. For example, the learning rate, tree complexity bag fraction, etc. Furthermore, it is crucial to identify optimal model parameters through cross-validation to reduce the overfitting of the model or to find the best performing model. I would suggest authors describe their methods related to cross-validation and hyper-tuning of model parameters in detail. Elith et al 2008 (A working guide to boosted regression trees) describe in detail the effects of various model parameters on the behavior of BRT models and describe the cross-validation method to find optimum model parameters. I would suggest authors run a cross-validation on the training dataset and test it over the remaining 25% of testing (validation/holdout dataset). This generally has been the standard in macro-ecological studies currently being presented (Han et al. 2015, Han et al. 2016). Inclusion of other model parameters such as family "Bernoulli" or "Poisson"?

2. Spatial resolution on data: The details about spatial resolution of tick occurrence data and predictor rasters used are lacking. It is not completely clear if tick occurrence was georeferenced to a prefecture-level (as shown in Figure 1) or it presented independent geolocations for each detection/occurrence. That also speaks towards the model family implemented (Poisson or Bernoulli). See the previous comment.

3. Predictions on counties with no prior history and accounting for sampling bias: Another important concern I have is with respect to accounting for spatial bias in sampling. Historical surveillance datasets are inherently biased. For example, regions with larger human presence are generally sampled more frequently than other regions. I think accounting for spatial sampling efforts in modeling either through experimental design and/or through the use of predictor that accounts for sampling effort. Refer to Allen et al. 2017 Nature Communications, for some ideas. Furthermore, authors use data from counties with at least one detection of tick species for training purposes but use all counties in China for Prediction purposes. This exacerbates sampling bias in model predictions. It will be ideal to restrict predictions to ecologically similar bio-geographical regions related to the ecology of tick species being modeled. Data collected by authors could greatly help in deciding how to restrict niche model predictions.

4. Comparison with other niche models for tick distributions: Numerous niche models for ticks are already present. VectorMap <http://vectormap.si.edu/dataportal.htm> compiles a comprehensive data based on for ticks and mosquitoes, niche models developed with similar mythologies. I would suggest authors compare models presented here with models developed by authors.

Without completely understanding the model behavior it is hard to assess model results related to the ecological niche of ticks and human cases. But the following are some minor edits and suggestions.

Other comments:

1. Please include line numbers and page numbers for easy review

Introduction:

2. "The family of pathogenic organisms harbored by ticks": viruses, bacteria are not taxonomic families organisms.

3. "Such expansion had caused severe emerging threat to human and animal health due to its capacity in transmitting life-threatening tick-borne agents such as severe fever with thrombocytopenia syndrome (SFTS) virus (SFTSV), spotted fever group rickettsiae, and *A. phagocytophilum*." This is a confusing sentence. Please rewrite it.

Results:

1. Change: "6954 unique records" to "6,954 unique records". Similarly, please include commas after every three decimal places in a number of four or more digits.

2. "The most widely distributed genus is" to "The most widely distributed genus was"

3. I understand that describing tick distributions based on the number of counties they were found in is critically important from management and policy perspective. But authors should also describe the distribution pertaining to their ecology. Which ticks were found diverse landscapes, various altitudes, and landcover or bio-geographic regions of China? The authors describe that briefly in results sections around figure 1, but more details will be welcome.

4. "100 models per set". What are the sets? If I understand well, they are 100 models for each tick species.

5. "Temperature seasonality and mean temperature in the driest quarter were the two most important drivers, contributing $\geq 5\%$ to the ensemble of models for 14 and 12 tick species, respectively, followed by altitude contributing $\geq 5\%$ to the models for nine tick species (Supplementary Tables 3–7).": I would recommend summarizing results in this paragraph a figure or some sort of visualization so that it is more easily accessible to readers.

6. Figure 2: include a label to the legend. If I understand it correctly, the colors in the heatmap are relative contributions of features.

7. What hierarchical clustering method was implemented (clustering of ecological features)?

Supplementary material

Page 9: "In the first stage, the split-and-fitting step was repeated for ten times to screen important predictors."

Spelling of important

Pranav Pandit,
BVSc & AH, MPVM, PhD
UC Davis

Reviewer #3:

Remarks to the Author:

This manuscript brings together many very important and relevant datasets ... data on tick distributions, pathogen detections, and human cases. In that sense, bravo! I love the potential for integration and cross-linking.

The problems that I noticed, however, were many:

1. There are many misspellings (e.g., "current filed surveillance" should be "current field surveillance"). I understand that the authors are working in a second language, and I "get it," about how hard that is. However, they also had many misspellings of scientific names, which are a universal language, so misspelling them is not excusable!
2. The color scheme in Figure 1 is terrible ... putting a pinkish on a green background is not colorblindness-friendly. See Color Brewer <http://colorbrewer2.org/> for helpful guidance.
3. "The BRT model sets (100 models per set) for the 19 predominant tick species all showed highly accurate prediction with the average area-under-curve (AUC) ranging from 0.83 to 0.97 (Table 1)." AUC is well known not to be appropriate for niche modeling applications such as those carried out in this study. See Lobo, J. M., A. Jiménez-Valverde, and R. Real. 2008. AUC: A misleading measure of the performance of predictive distribution models. *Global Ecology and Biogeography* 17:145-151. Also Peterson, A. T., M. Papeş, and J. Soberón. 2008. Rethinking receiver operating characteristic analysis applications in ecological niche modelling. *Ecological Modelling* 213:63-72.
4. Table 1 - write out genus names on first mention in the first column.
5. Please use "elevation" instead of "altitude." Elevation refers to vertical displacement from sea level, whereas altitude refers to vertical displacement from the ground.
6. "The model-predicted high risk areas of the 19 tick species were much more extensive than been observed" ... this could mean incomplete sampling and broader ranges than has been appreciated, OR it could mean underspecified models that overpredict the distributional area of the species. You should discuss these possibilities.
7. "The rapid spread of TBDs in the recent decades poses severe threats to human health and highlights the need to identify the natural foci, underlying socio-environmental drivers, and potential high risk zones for both vectors and pathogens." Have TBDs SPREAD rapidly in recent decades, or have they always been this broad, and we are just starting to appreciate it? Make this clear, one way or the other, and maybe even look at your data to document the spread process.
8. "The family of Ixodes, mainly comprising..." Ixodes is a genus. Also, family names of ticks are not italicized, whereas family names of viruses are italicized.
9. The description of the methods for the ecological niche modeling is entirely inadequate. Methods descriptions for such analyses generally extend 1-1.5 pages, not 1 paragraph! In particular, you need to justify how you managed the county-level spatial referencing. It is not easy, and I am not at all clear as to how you did it.

10. "The raw data that support the findings of this study are available from the corresponding author upon reasonable request." This is not appropriate. What is reasonable? What happens if the authors do not respond? Make the data OPEN!

In sum, what an interesting paper, but it is currently fraught with problems. I hope that these comments are useful, but the revision will have to be massive.

Responses to Reviewer 1:

1. This comprehensive work is built on a review of tick and tick-borne pathogen occurrence studies conducted in China (1950-2018). It summarizes the distribution of 124 tick species (at the county-level), and the associated pathogens carried by 56 tick species. In general, the manuscript misses providing a current understanding of ticks' biogeography and taxonomy in China as provided in several recent relevant review studies and comparing itself with them to update such an understanding. This makes it look as if the authors lacked the scientific ground they stood on or that they were avoiding these studies because of the overlapping outcomes. These studies are:

- (a) Zhang et al. (2018, Archives of Insect Biochemistry and Physiology, DOI: 10.1002/arch.21544) conducted a mini-review on the distribution of 125 tick species as well as their corrected names in China at the province level;
- (b) Zhang et al. (2019, Scientific Data, DOI: 10.1038/s41597-019-0115-5) reviewed the location records of tick in China reported 1960-2017 and published point-based, open access spatial data on the distribution of 123 tick species ;
- (c) Yu et al. (2015, Parasites & Vectors DOI: 10.1186/s13071-014-0628-x) reviewed articles and provided a table of a range of potential disease causative agents in 52 tick species found in China.

Thus, the existing evidence has already built a sound, comprehensive and reproducible outcome regarding the current distribution of ticks and the pathogens in China. The originality this manuscript may add on them should be based on an insightful analysis on the differences from or the corrections to the existing knowledge. My recommendation is to compare with these studies, and some others, and update the understanding of the distribution and taxonomy of ticks and the pathogens they harbor in China.

[Response] We are grateful for the reviewer's valuable comments. We have read carefully the suggested papers and have added the following in the Introduction to give readers a better idea about previous evidence and the added value of our work: "In China, the growing awareness of emerging tick-borne pathogens has inspired an enhanced investigation on ticks and TBDs in recent years^{4,14}. One study compiled a data set with regard to tick distribution and diversity up to the county level in China from peer-reviewed literature published between 1960 and 2017¹⁵. Another study reviewed the geographic distribution of tick species at the province level together with the diversity and specificity of animal hosts of ticks¹⁶. Yu et al. reviewed the association between pathogenic microorganisms and tick vectors throughout China based on the literature up to 2014¹⁷. However, none of the studies provided high-resolution spatial distribution of tick-borne pathogens, nor did they investigate systematically ecological niches of either major tick species or prevalent tick-borne pathogens. Here we conduct an up-to-date review on the spatial distribution of predominant tick species, tick-borne agents and human cases of TBDs in China, based on which we build predictive models to assess the contributions of relevant socioenvironmental factors to the ecological

suitability of selected ticks and tick-borne agents, and map model-projected risks to inform future surveillance and control efforts.” (Page 3-4, Lines 66–81)

In addition, we have compared our records with the data provided in these papers, especially the county-level data in Zhang et al., and updated our analyses on 31 tick species (4 *Demacentor* species, 12 *Haemaphysalis* species, 5 *Ixodes* species, 5 *Rhipicephalus* species and 2 *Ornithodoros* species) related to 1,134 counties. Accordingly, we have updated Figures 1, 2, and 5 and Tables 1 and 2 in the main text, as well as all tables and figures except Supplementary Fig. 9 in the Supplementary Information. There is no, however, qualitative change in these results and our conclusions.

In the discussion, we compared our results with these recent reviews and further emphasized the new contributions of our work (Page 21, Lines 333-340; Page 22, Lines 341-344) : “We assembled the most comprehensive, if not all, records of tick species and tick-borne pathogens in China that covers a time span of 70 years. We reported detection locations of some tick species not covered in existing compiled data sets¹⁶, e.g., *H. longicornis* in Xinjiang and *I. persulcatus* in Jiangsu (Supplementary Figs. 2-3). Our cross-tabulation of tick species and associated pathogens is also more complete than previous studies, e.g., listing ten known pathogens detected in *H. japonica* and nine tick species harboring SFTSV, compared to only two and one, respectively, in a previous review¹⁷. ”

In addition to ticks and pathogens, we also mapped the spatial distributions of human cases of tick-borne disease at the county level in China, unprecedented by any other study to our knowledge.

2. One of the main results of this manuscript is on the mapping and prediction of the potential distribution of SFTSV and TBE in China using the boosted regression tree (BRT) model. These parts of the work were based on previous studies published by the same authors or authors from the same institution. Fang et al. (2015, The Lancet Infectious Disease, DOI: 10.1016/S1473-3099(15)00177-2), reviewed existing and emerging tick-borne infections in mainland China, with the distribution of SFTSV being one of the foci of the paper. Sun et al. (2017, Ticks and Tick-borne Diseases, DOI: 10.1016/j.ttbdis.2017.04.009) reviewed and accessed the evidence on the distribution of ticks and TBE incidence across China, and mapped the potential distribution of TBE using the same BRT approach. Sun’s results on the predicted TBE map looks very similar to the one represented here in the manuscript. As a comparison with these previous works was not conducted either, it seems that only minor updates have been made compared with these studies published five or six years ago. My recommendation is to explicitly explain/discuss the new/different findings of this work.

[Response] This is a great suggestion. We have now discussed the similarities and differences between the current study and previous studies (page 21, lines 333-340; page 24, lines 395-399). Briefly, Fang et al. 2015 in the Lancet Infectious Diseases does not contain any ecological modeling results. Our previous modeling papers,

Miao et al. (2020, CID) on SFTS and Sun et al. (2017, Ticks & TBD) on TBE focused on the ecology of human diseases, whereas the current work focuses on the ecology of pathogens detected in both human cases and ticks. For this reason, ecological predictors also differ to some degree. For example, more demographics and agricultural variables were included in Miao et al. (2020) but not in the current work. In addition, the data used in this work are much more up to date than Fang et al. (2015) and Sun et al. (2017). Miao et al. (2020) also used the SFTS case data as of 2018, but the methodology differs fundamentally from the current study, e.g., the outcome there was yearly incidence of SFTS during 2010-2018, whereas the outcome in this study is the cumulative detection results up to 2018. The selection of control sites also differs greatly.

3. The design of the BRT modeling approach may need a better explanation and justification. (a) Rationales behind the selection of predictors and data processing steps are missing (Appendix pp.6-7; Supplementary Table 9). It reads that “climatic data were collected from 2006 surveillance stations in mainland China, covering 68.7% counties”. Then, how many counties with tick presence did not have surveillance stations?

[Response] We have expanded the section “Data on socioenvironmental and ecoclimatic factors” in the Supplementary Information (pages 5-7) to elaborate the rationales behind the selection of predictors and data processing steps. Details about the calculations of the ecoclimatic variables were given in the referenced paper (O’Donnell and Ignizio, 2012), but Supplementary Table 9 also contains brief description about the calculations. We have also rewritten the corresponding paragraph in Methods of the main text to better reflect the rationales (pages 26, line 452-456; page 27, lines 465-468). A total of 326 (28.8%) counties with tick presence did not have surveillance stations, for which the mean values of the nearest five surveillance stations were used as a proxy for their meteorological variables (Supplementary Materials and Methods, page 6, line 58-60).

Besides, why to include such a set of the 19 eco-climatic variables?

[Response] As mentioned above and the revised main text (pages 26-27) and the Supplementary Information (pages 5-6), these 19 ecoclimatic variables better capture seasonal trends of different species related to their physiological constraints than traditional climatic variables and have been widely used in ecological studies as well as recommended by the U.S. Geological Survey.

Why to only include climatic data between 1981-2010, presumably the post-2010 data records on tick and tick-borne pathogens were more abundant (Zhang et al., 2019)?

[Response] Per your suggestion, we have collected and processed the post-2010 climatic data at the county level, re-built the BRT models and updated the results for the 19 predominant tick species and the two tick-borne pathogens.

Why to use the land cover dataset of 2005, provided that there has been a continuous rapid land cover change in China thereafter?

[Response] The land cover data were updated about every 5-10 years in China, and the land cover categories and classification criteria often changed, making data sets from different years not directly comparable or combinable. Our tick records span over the past five decades. Although many tick records were collected post 2010, the establishment of their habitat likely has evolved much longer. Therefore, we used the 2005 land cover data for modeling the distribution of tick species. We have updated modeling results for the distribution of SFTSV and TBEV using the more recent 2015 land cover data, because a large portion of the data were recorded over the recent decade, in particular for SFTSV (after 2010). We have explained these reasons in the Supplementary Information (page 6) as well as in the Methods of the main text (pages 26, lines 457-463). We have acknowledged the issue of rapid land cover change in the recent decade as a limitation of our study in the revised manuscript. (page 25, lines 426-429)

Why to include rural population (proportion and density) as predictors when modeling the distribution of ticks and pathogens?

[Response] We included the rural population (proportion and density) in the models because patients of tick-borne diseases were mostly rural residents and domestic animals in rural areas serve as feeding hosts for many tick species. However, both variables have negligible contributions to the detection of ticks or tick-borne pathogens and were excluded from the final models. We explained the rationale in the Methods section (page 26, lines 453-457).

(b) While conducting the case-control design to select data points for BRT modeling, it reads that “those with surveys conducted but without detection of the tick species were considered as ‘control’ sites” (Appendix pp.7-8). Since the sample size matters, it would be better to provide at least the number of ‘control’ sites when modeling each of the 19 tick species. This could enhance readers’ confidence in accepting these predictions.

[Response] We agree with you and have provided the number of control sites for each tick species in the revised Supplementary Table 1.

(c) Finally, in the abstract, it reads that “major tick species are demonstrated to have much more extensive suitable habitats than what have been observed”. This is somewhat not surprising because every species distribution model will produce predictions like that.

[Response] We agree with you, and have rewritten this part of the abstract as: “The model-predicted suitable habitats for the 19 tick species are extensive, 15–453% larger in size than the geographic areas where these species have been observed. Tick species that are severely under-detected but harboring pathogens of imminent threats to public health should be prioritized for field surveillance.”

Overall, I think the manuscript has aimed too high to compile all the relevant knowledge on ticks and tick-borne disease in China, making it losses focuses and lacks clear/key take-home messages. I do appreciate the authors’ effort on conducting such an organized and comprehensive study like this. However, I would regard this work more suitable for specialized journals, such as International Journal for Parasitology, Parasites & Vectors, and Ticks & Tick-borne Diseases.

[Response] Thank you for the suggestion. While the scope of our manuscript is indeed wide, it connects the three inherently linked components of tick-borne diseases, the tick vectors, the pathogens, and human patients. Leaving any component out of this review will make it incomplete and regretful. Given that tick-borne diseases affect more and more people globally in recent years, we believe the wide scope and detailed data presented here are actually important strengths of the study. We also believe Nature Communications is the right venue to disseminate the results to the broad readers to raise their awareness about tick-borne diseases and underlying drivers. Our data shared via Nature Communications will hopefully stimulate more research efforts in this field.

References

- 1 Zhang, Y. K., Zhang, X. Y. & Liu, J. Z. Ticks (Acari: Ixodoidea) in China: Geographical distribution, host diversity, and specificity. *Arch Insect Biochem Physiol* 102, e21544 (2019). <https://doi.org/10.1002/arch.21544>
- 2 Zhang, G., Zheng, D., Tian, Y. & Li, S. A dataset of distribution and diversity of ticks in China. *Sci Data* 6, **105** (2019). <https://doi.org/10.1038/s41597-019-0115-5>
- 3 Yu, Z. et al. Tick-borne pathogens and the vector potential of ticks in China. *Parasit Vectors* 8, **24** (2015). <https://doi.org/10.1186/s13071-014-0628-x>

Responses to Reviewer 2:

In this study Zhao et al. has collected comprehensive data on tick species in China abundance and described their ecological niche using boosting regression trees. Along with developing models for the nineteen most common tick species found in China they also present a model describing the spatial risk of tick-borne pathogens specifically Tick-borne encephalitis virus (TBEV) and Severe Fever with thrombocytopenia syndrome virus (SFTSV). The detailed data presented here in the manuscript is one of the important strengths of the study. Keeping in mind the importance of ticks and tick-borne infections for public health the scope of the manuscript is relevant. Authors do a great job in describing the selection of predictors, case-control experimental set and use of randomly selected 75% data as the training set to test on the remaining 25% of the data. However, I have some concerns related to the validation of models presented in the study, especially hyper-tuning of boosted regression trees and validation through the cross validation-holdout method. Details of which are lacking the manuscript.

Major concerns related to analysis:

1. Cross-validation and optimum model parameters: It is not clear on how boosted regression tree (BRT) models were developed and what exactly was the outcome variable of interest and how the corresponding BRT model was developed. For example, if the model is used to predict presence and absence of (binary 1, 0) of tick species or abundance of tick species is unclear? The authors do not present the details of the BRT model parameters used for all the models developed here. For example, the learning rate, tree complexity bag fraction, etc. Furthermore, it is crucial to identify optimal model parameters through cross-validation to reduce the overfitting of the model or to find the best performing model. I would suggest authors describe their methods related to cross-validation and hyper-tuning of model parameters in detail. Elith et al 2008 (A working guide to boosted regression trees) describe in detail the effects of various model parameters on the behavior of BRT models and describe the cross-validation method to find optimum model parameters. I would suggest authors run a cross-validation on the training dataset and test it over the remaining 25% of testing (validation/holdout dataset). This generally has been the standard in macro-ecological studies currently being presented (Han et al. 2015, Han et al. 2016). Inclusion of other model parameters such as family “Bernoulli” or “Poisson”?

[Response] We have actually performed 10-fold cross-validation to find the optimum number of trees for each model, which was implemented using the `gbm.step` function in the R package *dismo*. We fixed the parameters learning rate (0.005), tree complexity (5) and bagging fraction (75%) at values proven to work well based on our previous ecological research. As both the data size (45 predictors) and the number of model runs ([19 ticks + 2 pathogens]×100) are pretty large, we cannot afford cross-validation for all the learning parameters. However, we have also tried a learning rate of 0.01 on a few major models such as for *Ha. longicornis* and for SFTSV, and found no difference in the results. We have

clarified cross-validation and sensitivity analysis in the Methods section (pages 27-28, lines 490-492) and in the Supplementary Information (page 8, 124-130; page 9, lines 139-142; page 10, lines 157-163).

2. Spatial resolution on data: The details about spatial resolution of tick occurrence data and predictor rasters used are lacking. It is not completely clear if tick occurrence was georeferenced to a prefecture-level (as shown in Figure 1) or it presented independent geolocations for each detection/occurrence. That also speaks towards the model family implemented (Poisson or Bernoulli). See the previous comment.

[Response] For the BRT models, the data of tick and pathogen occurrence as well as predictors were all at the county level. For maps of spatial distributions, tick species were all mapped at the county level (Supplementary Figs. 1-10), but pathogens were mapped at either the county level or prefecture level, depending on data availability (Supplementary Figs. 35-41). In Figure 4, patients infected by tick-borne pathogens (SFTSV and TBEV excluded) were plotted at either county, prefecture or province level, depending on available resolution. For SFTSV and TBEV, data of both pathogens and patients were all at the county level. In Fig. 1, we displayed tick species richness at the prefecture-level simply for the convenience of presentation. We have carefully checked methods description in Methods and in the Supplementary Information as well as all figure legends and table captions to make sure the spatial resolution is clearly stated.

3. Predictions on counties with no prior history and accounting for sampling bias: Another important concern I have is with respect to accounting for spatial bias in sampling. Historical surveillance datasets are inherently biased. For example, regions with larger human presence are generally sampled more frequently than other regions. I think accounting for spatial sampling efforts in modeling either through experimental design and/or through the use of predictor that accounts for sampling effort. Refer to Allen et al. 2017 Nature Communications, for some ideas. Furthermore, authors use data from counties with at least one detection of tick species for training purposes but use all counties in China for Prediction purposes. This exacerbates sampling bias in model predictions. It will be ideal to restrict predictions to ecologically similar bio-geographical regions related to the ecology of tick species being modeled. Data collected by authors could greatly help in deciding how to restrict niche model predictions.

[Response] We appreciate and agree with the reviewer's comments. As is well known, field surveillance rarely samples sites randomly and researchers more likely sample "high risk" places where they believe the ticks are more likely to be found. We did not try to find variables that may be predictive of being surveyed vs. not surveyed, because we do not have sufficient information about the sampling

design of historical surveys. This biased sampling will bias our analyses towards the null, i.e., we might not be able to detect some important contributing factors, if these factors were related to the sampling of survey sites. This bias, however, is less likely to cause false positive findings, i.e., factors that take no actual effect are flagged as important contributors. In addition, the surveyed 1,134 counties (39% of all counties in China) covered all bio-geographical regions of China (Supplementary Fig. 1), which implies a low likelihood of geographic bias. Finally, we sampled 75% of surveyed counties both with (cases) and without (controls) detection of tick species for training purpose, not just counties with at least one detection (Supplementary Information, page 9, lines 139-142). We discussed this issue of sampling bias as a limitation of this study in the revised manuscript (page 25, lines 419-425).

4. Comparison with other niche models for tick distributions: Numerous niche models for ticks are already present. VectorMap <http://vectormap.si.edu/dataportal.htm> compiles a comprehensive data based on for ticks and mosquitoes, niche models developed with similar mythologies. I would suggest authors compare models presented here with models developed by authors.

[Response] We thank the reviewer for bringing this interesting site to our attention. There are quite a few popular machine learning algorithms for ecological modeling, such as Boosted Regression Tree (BRT), Random Forest (RF), Support Vector Machine (SVM), Maximum Entropy (Maxent), etc. Among these, Maxent is more used for presence-only data. For case-control studies, BRT and RF are seen more frequently than others, and both are well developed and studied methods. In our previous work on modeling annual presence and incidence of SFTS patients in China (Miao et al., *Clinical Infectious Diseases*, 2020), we compared BRT, RF and SVM for predicting presence. BRT showed slightly better performance than the other two. In the current work, our focus is distribution and ecology of 19 tick species and 2 tick-borne pathogens. Due to the computational burden, we chose BRT based on our previous experience and found satisfactory performance. Another property of BRT more desirable to us is that BRT tends to shrink the contributions of unimportant predictors, whereas RF tends to give a more continuous spectrum of importance. We agree using multiple methods will ensure the robustness of the conclusions, but we think that would be more appropriate when the ecology of one or two species is to be thoroughly investigated, e.g., as in the example studies listed below. Such model comparison in our situation may make the paper too lengthy and less focused.

Freeman et al. Random forests and stochastic gradient boosting for predicting tree canopy cover: comparing tuning processes and model performance. *Canadian Journal of Forest Research*. 2016; 46: 323-339, <https://doi.org/10.1139/cjfr-2014-0562>

Yang et al. Comparison of boosted regression tree and random forest models for mapping topsoil organic carbon concentration in an alpine ecosystem. Ecological Indicators. 2016;60: 870-878. <https://doi.org/10.1016/j.ecolind.2015.08.036>

Without completely understanding the model behavior it is hard to assess model results related to the ecological niche of ticks and human cases. But the following are some minor edits and suggestions.

Other comments:

1. Please include line numbers and page numbers for easy review

Introduction:

[Response] We have included both page numbers and line numbers in the revised manuscript.

2. “The family of pathogenic organisms harbored by ticks”: viruses, bacteria are not taxonomic families organisms.

[Response] Thank you for pointing this out. We have rephrased this sentence as “Pathogenic organisms harbored by ticks mainly encompasses viruses, bacteria (in particular rickettsiae and spirochaetes), protozoa, and helminth”. (page 3, lines 49-51)

3. “Such expansion had caused severe emerging threat to human and animal health due to its capacity in transmitting life-threatening tick-borne agents such as severe fever with thrombocytopenia syndrome (SFTS) virus (SFTSV), spotted fever group rickettsiae, and *A. phagocytophilum*.”: This is a confusing sentence. Please rewrite it.

[Response] We have revised this sentence as “The expansion of *Ha. longicornis* has raised public health and animal health concerns due to its capability if transmitting tick-borne agents, e.g., severe fever with thrombocytopenia syndrome virus (SFTSV), spotted fever group *rickettsiae*, and *A. phagocytophilum*”. (page 3, lines 56-59)

Results:

1. Change: “6954 unique records” to “6,954 unique records”. Similarly, please include commas after every three decimal places in a number of four or more digits.

[Response] We have checked the main text and the Supplementary Information to ensure uniform use of this number format.

2. “The most widely distributed genus is” to “The most widely distributed genus was”

[Response] We have made changes as suggested.

3. I understand that describing tick distributions based on the number of counties they were found in is critically important from management and policy perspective. But authors should also describe the distribution pertaining to their ecology. Which ticks were found diverse landscapes, various altitudes, and landcover or bio-geographic regions of China? The authors describe that briefly in results sections around figure 1, but more details will be welcome.

[Response] Most descriptions about the ecological environments and landscapes inhabited by the tick species and their clusters were given in the subsection titled “Ecological clustering of tick species” of the Results section. We like your suggestion of describing biogeographic zones. We have added boundaries of biogeographic zones to Fig. 2 and also described which clusters fall in which biogeographic zones in that subsection (pages 9-11, lines 150-179).

4. “100 models per set”. What are the sets? If I understand well, they are 100 models for each tick species.

[Response] For each of the 19 predominant tick species, we resample the data 100 times, each time with 75% data for model training and 25% for testing; therefore, each tick species is associated with a set of 100 BRT models. This is detailed in the Methods section (pages 27-28, lines 486-497) as well as in the Supplementary Information (pages 9-10, lines 136-157). To avoid confusion, we have removed this sentence as the technical information has been given in the Methods.

5. “Temperature seasonality and mean temperature in the driest quarter were the two most important drivers, contributing $\geq 5\%$ to the ensemble of models for 14 and 12 tick species, respectively, followed by altitude contributing $\geq 5\%$ to the models for nine tick species (Supplementary Tables 3–7).”: I would recommend summarizing results in this paragraph a figure or some sort of visualization so that it is more easily accessible to readers.

[Response] We are now referring readers to Fig. 2F which is a natural visualization of relative contributions (represented by colors) of important drivers

for all the 19 predominant tick species (page 9, lines 127-131).

6. Figure 2: include a label to the legend. If I understand it correctly, the colors in the heatmap are relative contributions of features.

[Response] Yes, your interpretation is correct. We have added labels to the legend of Fig. 2 to clarify the meanings of two quantities: “Two of the three quantities were displayed in panel F to indicate the possible level of ecological suitability: relative contributions (colors in ascending order from yellow to red) and standardized median value of the predictor (numbers in the heatmap) among counties with tick occurrence (numbers 1–4 indicate the position of this median in reference to the quartiles of this predictor among all counties)”.

7. What hierarchical clustering method was implemented (clustering of ecological features)?

[Response] This clustering method is called the weighted-average linkage method implemented in the statistical software Stata (Hamilton, Statistics with STATA. Boston, Cengage Learning, 2009). Details can be found in the Supplementary Materials and Methods section in the Supplementary Information (pages 10-11).

Supplementary material

Page 9: “In the first stage, the split-and-fitting step was repeated for ten times to screen important predictors.”

Spelling of important

[Response] Thank you for pointing this typo out. We have carefully checked the revised manuscript and Supplementary Materials and Methods for spelling errors.

Responses to Reviewer 3:

This manuscript brings together many very important and relevant datasets ... data on tick distributions, pathogen detections, and human cases. In that sense, bravo! I love the potential for integration and cross-linking.

The problems that I noticed, however, were many:

1. There are many misspellings (e.g., "current filed surveillance" should be "current field surveillance"). I understand that the authors are working in a second language, and I "get it," about how hard that is. However, they also had many misspellings of scientific names, which are a universal language, so misspelling them is not excusable!

[Response] We apologize for the misspellings and have checked the manuscript throughout to remove spelling and grammar mistakes.

2. The color scheme in Figure 1 is terrible ... putting a pinkish on a green background is not colorblindness-friendly. See Color Brewer <http://colorbrewer2.org/> for helpful guidance.

[Response] Thanks. We have adjusted the color scheme in Fig. 1.

3. "The BRT model sets (100 models per set) for the 19 predominant tick species all showed highly accurate prediction with the average area-under-curve (AUC) ranging from 0.83 to 0.97 (Table 1)." AUC is well known not to be appropriate for niche modeling applications such as those carried out in this study. See Lobo, J. M., A. Jiménez-Valverde, and R. Real. 2008. AUC: A misleading measure of the performance of predictive distribution models. *Global Ecology and Biogeography* 17:145-151. Also Peterson, A. T., M. Papeş, and J. Soberón. 2008. Rethinking receiver operating characteristic analysis applications in ecological niche modelling. *Ecological Modelling* 213:63-72.

[Response] We agree that AUC is not ideal and can be misleading for comparing models. We thank the reviewer for suggesting the two AUC-related papers and read them with interest. Lobo et al. (2008) mainly talks about limitations of AUC. Peterson et al. (2008) focuses on partial-area AUC which is suitable for comparing models with inherently different omission error ranges or when user prespecify a tolerance level of omission error, particularly for models without controls (i.e., presence-only data). Our data and study design are different. The ROC curves produced by our models for tick species and for the pathogens are all continuous, i.e., the BRT model has an omission error range of 0-1 rather than only a partial range. Secondly, we are not using AUC for comparison between different models. In addition, our study is a case-control design, where controls were based on a large-scale survey for ticks. Finally, the choice of a tolerance error level is subjective, and we do not have any prior belief. For these reasons, we think our use of AUC is justifiable. On the other hand, we agree that AUC should not be over-interpreted especially in ecology studies. We have added the following in Discussion to acknowledge the limitation of AUC (Page 25, Lines 430-433): "while the AUC values are high for all the models we fitted, AUC does not necessarily reflect the goodness of fit and could be misleading when the absence data are

associated with high uncertainty¹. Such uncertainty exists as most surveys are cross-sectional”.

4. Table 1 - write out genus names on first mention in the first column.

[Response] We appreciate your suggestion and have included the genus name in the first column in the revised Table 1.

5. Please use "elevation" instead of "altitude." Elevation refers to vertical displacement from sea level, whereas altitude refers to vertical displacement from the ground.

[Response] We have replaced “elevation” by “altitude” throughout the manuscript.

6. "The model-predicted high risk areas of the 19 tick species were much more extensive than been observed" ... this could mean incomplete sampling and broader ranges than has been appreciated, OR it could mean underspecified models that overpredict the distributional area of the species. You should discuss these possibilities.

[Response] We agree with you and have discussed at the end of the first paragraph in Discussion: “However, it is also possible that our models were underspecified and thus overestimated the scopes.” (page 22, lines 344-345).

7. "The rapid spread of TBDs in the recent decades poses severe threats to human health and highlights the need to identify the natural foci, underlying socio-environmental drivers, and potential high risk zones for both vectors and pathogens." Have TBDs SPREAD rapidly in recent decades, or have they always been this broad, and we are just starting to appreciate it? Make this clear, one way or the other, and maybe even look at your data to document the spread process.

[Response] In response to the reviewer’s suggestion, we have removed this sentence from our manuscript as spread process of TBDs has not been explored in this study.

8. "The family of Ixodes, mainly comprising..." Ixodes is a genus. Also, family names of ticks are not italicized, whereas family names of viruses are italicized.

[Response] We have corrected it. We have checked family names of ticks throughout the manuscript to make sure they are not italicized.

9. The description of the methods for the ecological niche modeling is entirely inadequate. Methods descriptions for such analyses generally extend 1-1.5 pages, not 1 paragraph! In particular, you need to justify how you managed the county-level spatial referencing. It is not easy, and I am not at all clear as to how you did it.

[Response] We have provided more details about ecological modeling in the Methods section (pages 27-28, lines 480-513). Further details can be found in the Supplementary Information (pages 7-10). Ticks and tick-borne pathogens were mapped at the county level by geo-referencing the location of each record. When county information is not available, prefecture or province are used for location. Human cases of TBDs were individually mapped at the county, prefecture or province level in a similar way. In total 78.8%, 63.3% and 60.6% of ticks, tick-borne pathogens and TBD were geo-referenced at the county level. We have added a short paragraph about geo-referencing in the Methods section (pages 27, lines 473-478). Please also see our response to Reviewer 2's second question.

10. "The raw data that support the findings of this study are available from the corresponding author upon reasonable request." This is not appropriate. What is reasonable? What happens if the authors do not respond? Make the data OPEN!

[Response] We have stated in the revised main text that we will make the raw data openly downloadable.

In sum, what an interesting paper, but it is currently fraught with problems. I hope that these comments are useful, but the revision will have to be massive.

References

- 1 Jorge M. Lobo, A. J.-V., Raimundo Real. AUC:a misleading measure of the performance of predictive distribution models. *Global Ecol. Biogeogr.*, 145–151 (2008). <https://doi.org/10.1111/j.1466-8238.2007.00358.x>

Reviewers' Comments:

Reviewer #1:

Remarks to the Author:

I think the revised manuscript has improved substantially, especially on the rationale behind and clarity of methodology. My comments are:

One of my previous suggestions was to point out the differences between the results of this study and the authors' previous works on mapping and prediction of the distribution of SFTSV and TBE in China. The authors responded mostly on the different data and methods used. However, it may be more interesting to explain the differences in their findings. Do the county-level geographical patterns of SFTSV and TBE differ significantly from the previous findings? Why?

I think it'd be a huge contribution to the field if this study's data on ticks, tick-borne pathogens and disease incidence would be made publicly available.

Reviewer #2:

Remarks to the Author:

There is a significant improvement in the manuscript, especially the way modeling procedures are described. However, I still have a few major concerns that authors need to improve upon. I still cannot find results related to model performance especially when trained models were tested /validated against the testing data. The AUC values presented in the supplementary table 1 seem to like are for the cross-validation AUC of the model. While testing AUC values are missing. Generally, discrepancies in training and testing AUC values helps identify if models are overfitted or not and if it makes sense to use them on completely new data (in this case counties without surveillance). I would like to see these results mentioned in the manuscript.

Authors still are not able to address issues related to sampling bias. While I agree that sampling bias will bias analysis towards null especially for factor contributions, we still can't estimate by how much. Further, this also means that predictions towards the risk for un-surveyed counties are wrong by some margins. I would strongly suggest authors include a predictive factor that represents the sampling efforts and I can suggest some avenues for further exploration. 1. Authors can see if any studies have already predicted spatial biases in sampling for emerging infectious diseases that they extract data (summarized or raw) and can use as a proxy for sampling bias. 2. Authors can try to generate simple predictive factors such as years since the first survey, or the number of surveys that have been conducted for each county. Some innovative thinking is crucial here, otherwise, I strongly feel that authors are missing a big effect (predictive factor) that has been proven important in many studies until now (Albery et al. 2020; Allen et al. 2017; Han et al. 2020; Han et al. 2016; Han et al. 2015; Johnson et al. 2020; Pandit et al. 2018).

The manuscript brings in crucial data on ticks and their ecology for the whole of China, which is a significant addition to the knowledge, but interpretations and conclusions from models need some bolstering especially due to major concerns that I have for the modeling procedures.

Albery GF, Eskew EA, Ross N, and Olival KJ. 2020. Predicting the global mammalian viral sharing network using phylogeography. *Nat Commun* 11:2260. [10.1038/s41467-020-16153-4](https://doi.org/10.1038/s41467-020-16153-4)

Allen T, Murray KA, Zambrana-Torrel C, Morse SS, Rondinini C, Di Marco M, Breit N, Olival KJ, and Daszak P. 2017. Global hotspots and correlates of emerging zoonotic diseases. *Nat Commun* 8:1124. [10.1038/s41467-017-00923-8](https://doi.org/10.1038/s41467-017-00923-8)

Han BA, O'Regan SM, Paul Schmidt J, and Drake JM. 2020. Integrating data mining and transmission theory in the ecology of infectious diseases. *Ecol Lett*. [10.1111/ele.13520](https://doi.org/10.1111/ele.13520)

Han BA, Schmidt JP, Alexander LW, Bowden SE, Hayman DT, and Drake JM. 2016. Undiscovered Bat Hosts of Filoviruses. *PLoS Negl Trop Dis* 10:e0004815. [10.1371/journal.pntd.0004815](https://doi.org/10.1371/journal.pntd.0004815)

Han BA, Schmidt JP, Bowden SE, and Drake JM. 2015. Rodent reservoirs of future zoonotic diseases. *Proc Natl Acad Sci U S A* 112:7039-7044. 10.1073/pnas.1501598112

Johnson CK, Hitchens PL, Pandit PS, Rushmore J, Evans TS, Young CCW, and Doyle MM. 2020. Global shifts in mammalian population trends reveal key predictors of virus spillover risk. *Proc Biol Sci* 287:20192736. 10.1098/rspb.2019.2736

Pandit PS, Doyle MM, Smart KM, Young CCW, Drape GW, and Johnson CK. 2018. Predicting wildlife reservoirs and global vulnerability to zoonotic Flaviviruses. *Nat Commun* 9:5425. 10.1038/s41467-018-07896-2

Reviewer #3:

Remarks to the Author:

I do not see the authors' response as adequate at all ... true, they accept the typos that were pointed out, but they avoid responding to the substantive issues. Other reviewers pointed out, and I concur, that BRT models are an extreme option among SDM techniques in that they tend to overfit quite tightly to the available data in environmental space, and yet the authors make no move towards exploring alternative methods that do not overfit as much.

Similarly, among my own comments, I pointed out that ROC AUC methods are not appropriate for this sort of study, and offered two published papers (from a decade ago, I should point out!) as evidence of this problem. The authors responded with a disjointed and unintelligible response ... excerpted as:

"We agree that AUC is not ideal and can be misleading for comparing models... Our data and study design are different. The ROC curves produced by our models for tick species and for the pathogens are all continuous, i.e., the BRT model has an omission error range of 0-1 rather than only a partial range." This is not AT ALL different. In fact, their data and study design are entirely typical, and not at all different.

"Secondly, we are not using AUC for comparison between different models." Be that as it may, the two papers that I provided to the authors point out that AUC is inappropriate for the entire enterprise, and not just for model comparisons. The method is rife with assumptions that are not met, and that render it not appropriate for this toolset.

"In addition, our study is a case-control design, where controls were based on a large-scale survey for ticks." Ummm, a case control study is defined as one that compares objects that have a quality (e.g., presence of a tick species) with objects that do not have it (controls), yet the authors do not comprehend that they lack these negative/control data. Their "absence" data are non-detections of tick species at sites in counties across China, which does NOT constitute absence, and therefore their study is NOT a case-control study.

"Finally, the choice of a tolerance error level is subjective, and we do not have any prior belief." The authors could and should be honest with themselves that their input data are rife with error, not out of any fault of their own. That is, they are using occurrences in counties across China as their presence data. However, they will have had to use an average environmental value across a county, or a value at the county centroid, to represent the environments, whereas the tick species may have been recorded in the county under conditions and circumstances that may deviate rather dramatically from the average or centroid value. That is, in effect, error, and the authors are neglecting it!

"For these reasons, we think our use of AUC is justifiable." I disagree!

Finally, I will point out that even the simple edits that I suggested were not followed ... e.g., on lines 89-92, scientific names are not italicized, and generic names are misspelled ("Dermacentors").

As such, I conclude that this revision is entirely inadequate.

Responses to the reviewers:

Reviewer #1 (Remarks to the Author):

(1) I think the revised manuscript has improved substantially, especially on the rationale behind and clarity of methodology. My comments are:

One of my previous suggestions was to point out the differences between the results of this study and the authors' previous works on mapping and prediction of the distribution of SFTSV and TBE in China. The authors responded mostly on the different data and methods used. However, it may be more interesting to explain the differences in their findings. Do the county-level geographical patterns of SFTSV and TBE differ significantly from the previous findings? Why?

[Response] We agree with your suggestions and provide more specific comparisons in the geographic distribution and affected population size between current results and previous studies. In addition, we plot figures (supplementary figures 45 and 46) to show the differences visually. We also described the differences in ecological drivers. Please see the 6th paragraph in the revised Discussion section (page 24, lines 397-414) and the revised Supplementary Information (pages 61-62).

(2) I think it'd be a huge contribution to the field if this study's data on ticks, tick-borne pathogens and disease incidence would be made publicly available.

[Response] Thank you. We will make our data publicly available. Once the paper is accepted, we will insert a data-sharing statement and a download link.

Reviewer #2 (Remarks to the Author):

(1) There is a significant improvement in the manuscript, especially the way modeling procedures are described. However, I still have a few major concerns that authors need to improve upon.

I still cannot find results related to model performance especially when trained models were tested /validated against the testing data. The AUC values presented in the supplementary table 1 seem to like are for the cross-validation AUC of the model. While testing AUC values are missing. Generally, discrepancies in training and testing AUC values helps identify if models are overfitted or not and if it makes sense to use them on completely new data (in this case counties without surveillance). I would like to see these results mentioned in the manuscript.

[Response] We appreciate your comments and suggestions. We now provide both training and testing AUC values for the tick models in the revised Supplementary Information (Supplementary Tables 3–7) as well as for the pathogen models (Table 2), which clearly indicate the models are not overfitted.

(2) Authors still are not able to address issues related to sampling bias. While I agree that sampling bias will bias analysis towards null especially for factor contributions, we still can't estimate by how much. Further, this also means that predictions towards the risk for un-surveyed counties are wrong by some margins. I would strongly suggest authors include a predictive factor that represents the sampling efforts and I can suggest some avenues for further exploration. 1. Authors can see if any studies have already predicted spatial biases in sampling for emerging infectious diseases that they extract data (summarized or raw) and can use as a proxy for sampling bias. 2. Authors can try to generate simple predictive factors such as years since the first survey, or the number of surveys that have been conducted for each county. Some innovative thinking is crucial here, otherwise, I strongly feel that authors are missing a big effect (predictive factor) that has been proven important in many studies until now (Albery et al. 2020; Allen et al. 2017; Han et al. 2020; Han et al. 2016; Han et al. 2015; Johnson et al. 2020; Pandit et al. 2018).

The manuscript brings in crucial data on ticks and their ecology for the whole of China, which is a significant addition to the knowledge, but interpretations and conclusions from models need some bolstering especially due to major concerns that I have for the modeling procedures.

Albery GF, Eskew EA, Ross N, and Olival KJ. 2020. Predicting the global mammalian viral sharing network using phylogeography. *Nat Commun* 11:2260. 10.1038/s41467-020-16153-4

Allen T, Murray KA, Zambrana-Torrel C, Morse SS, Rondinini C, Di Marco M, Breit N, Olival KJ, and Daszak P. 2017. Global hotspots and correlates of emerging zoonotic diseases. *Nat Commun* 8:1124. 10.1038/s41467-017-00923-8

Han BA, O'Regan SM, Paul Schmidt J, and Drake JM. 2020. Integrating data mining and transmission theory in the ecology of infectious diseases. *Ecol Lett*. 10.1111/ele.13520

Han BA, Schmidt JP, Alexander LW, Bowden SE, Hayman DT, and Drake JM. 2016. Undiscovered Bat Hosts of Filoviruses. *PLoS Negl Trop Dis* 10:e0004815. 10.1371/journal.pntd.0004815

Han BA, Schmidt JP, Bowden SE, and Drake JM. 2015. Rodent reservoirs of future zoonotic diseases. *Proc Natl Acad Sci U S A* 112:7039-7044. 10.1073/pnas.1501598112

Johnson CK, Hitchens PL, Pandit PS, Rushmore J, Evans TS, Young CCW, and

Doyle MM. 2020. Global shifts in mammalian population trends reveal key predictors of virus spillover risk. *Proc Biol Sci* 287:20192736. 10.1098/rspb.2019.2736
Pandit PS, Doyle MM, Smart KM, Young CCW, Drape GW, and Johnson CK. 2018. Predicting wildlife reservoirs and global vulnerability to zoonotic Flaviviruses. *Nat Commun* 9:5425. 10.1038/s41467-018-07896-2

[Response] We appreciate your suggestions and agree sampling bias should be addressed. We have read the suggested papers with interest. To address the sampling bias issue, we did the following. We first built a logistic regression model for the selection of tick survey sites with a variety of ecoclimatic and socioenvironmental variables as predictors. The response of this model is 1 for all tick-surveyed counties and 0 for other counties. The most predictive model was chosen by p-value < 0.05 using a backward model-selection procedure. The reciprocals of model-predicted sampling probabilities of all surveyed counties were then used as weights to re-fit the BRT models for the 19 major tick species. We explained this method in the methods section (pages 28–29, lines 518–525). We have updated all BRT results for the 19 major tick species in the revised manuscript and Supplementary Information. The results were similar to those shown in the previous version.

Reviewer #3 (Remarks to the Author):

I do not see the authors' response as adequate at all ... true, they accept the typos that were pointed out, but they avoid responding to the substantive issues. Other reviewers pointed out, and I concur, that BRT models are an extreme option among SDM techniques in that they tend to overfit quite tightly to the available data in environmental space, and yet the authors make no move towards exploring alternative methods that do not overfit as much.

Similarly, among my own comments, I pointed out that ROC AUC methods are not appropriate for this sort of study, and offered two published papers (from a decade ago, I should point out!) as evidence of this problem. The authors responded with a disjointed and unintelligible response ... excerpted as:

"We agree that AUC is not ideal and can be misleading for comparing models... Our data and study design are different. The ROC curves produced by our models for tick species and for the pathogens are all continuous, i.e., the BRT model has an omission error range of 0-1 rather than only a partial range." This is not AT ALL different. In fact, their data and study design are entirely typical, and not at all different.

"Secondly, we are not using AUC for comparison between different models." Be that as it may, the two papers that I provided to the authors point out that AUC is

inappropriate for the entire enterprise, and not just for model comparisons. The method is rife with assumptions that are not met, and that render it not appropriate for this toolset.

"In addition, our study is a case-control design, where controls were based on a large-scale survey for ticks." Ummm, a case control study is defined as one that compares objects that have a quality (e.g., presence of a tick species) with objects that do not have it (controls), yet the authors do not comprehend that they lack these negative/control data. Their "absence" data are non-detections of tick species at sites in counties across China, which does NOT constitute absence, and therefore their study is NOT a case-control study.

"Finally, the choice of a tolerance error level is subjective, and we do not have any prior belief." The authors could and should be honest with themselves that their input data are rife with error, not out of any fault of their own. That is, they are using occurrences in counties across China as their presence data. However, they will have had to use an average environmental value across a county, or a value at the county centroid, to represent the environments, whereas the tick species may have been recorded in the county under conditions and circumstances that may deviate rather dramatically from the average or centroid value. That is, in effect, error, and the authors are neglecting it!

"For these reasons, we think our use of AUC is justifiable." I disagree!

[Response] We appreciate your comments. We apologize for not being able to address your concerns in the first round of revision. We indeed have enjoyed reading the papers you recommended, especially the Peterson et al. paper which proposed the use of partial area AUC. In that paper, Peterson et al. stated two scenarios for which the partial area AUC should be considered, (1) unequal span of false positives (commission error) between models, and (2) niche modeling where negative controls (absence) are either not available or not meaningful. We agree that the "control" sites in our study not necessarily reflect real absence due to either incomplete survey or biased survey. For modeling the two pathogens, SFTSV and TBEV, we took extra caution by applying stringent criteria to the definition of a "control" county (Supplementary Information [SI], page 13). On the other hand, totally discarding control sites may also lead to serious loss of information. To balance between the two considerations, we now present both the AUC when the tolerance level (E) is set to 100% (full area AUC) and when E=20% (partial area AUC) for all BRT models for the 19 main tick species (SI, pages 72–77, Supplementary Tables 3–7) as well as for the two pathogens SFTSV and TBEV (Table 2). For partial area AUC, the horizontal axis is the total rate of positives rather than false positives. We presented the ratio of the partial AUC to the area under the random selection line (diagonal line) as suggested by Peterson et al. Most ratios for test data sets are near or over 1.5, indicating decent

predictive power of the models. We have modified the Methods section (page 28, lines 516–517) and the Results section (page 6, lines 120–121) in the main text as well as in the SI (page 11, lines 165–170) to reflect the use of partial area AUC.

Finally, I will point out that even the simple edits that I suggested were not followed ... e.g., on lines 89-92, scientific names are not italicized, and generic names are misspelled ("Dermacentors").

As such, I conclude that this revision is entirely inadequate.

[Response] Thank you for pointing this out. We have carefully checked the revised manuscript and SI for spelling errors.

Reviewers' Comments:

Reviewer #2:

Remarks to the Author:

The authors have satisfactorily addressed two of my major concerns and I do not have any additional comments.